# A non-archaeopterygid avialan theropod from the Late Jurassic of southern Germany

Oliver WM Rauhut[1,2,3†]*, Helmut Tischlinger[4], Christian Foth[5†]

[1]Staatliche naturwissenschaftliche Sammlungen Bayerns (SNSB), Bayerische Staatssammlung für Paläontologie und Geologie, München, Germany; [2]Department for Earth and Environmental Sciences, Palaeontology and Geobiology, Ludwig-Maximilians-Universität, München, Germany; [3]GeoBioCenter, Ludwig-Maximilians-Universität, München, Germany; [4]Tannenweg 16, Stammham, Germany; [5]Department of Geosciences, Université de Fribourg, Fribourg, Switzerland

**Abstract** The Late Jurassic 'Solnhofen Limestones' are famous for their exceptionally preserved fossils, including the urvogel *Archaeopteryx*, which has played a pivotal role in the discussion of bird origins. Here we describe a new, non-archaeopterygid avialan from the Lower Tithonian Mörnsheim Formation of the Solnhofen Archipelago, *Alcmonavis poeschli* gen. et sp. nov. Represented by a right wing, *Alcmonavis* shows several derived characters, including a pronounced attachment for the pectoralis muscle, a pronounced tuberculum bicipitale radii, and a robust second manual digit, indicating that it is a more derived avialan than *Archaeopteryx*. Several modifications, especially in muscle attachments of muscles that in modern birds are related to the downstroke of the wing, indicate an increased adaptation of the forelimb for active flapping flight in the early evolution of birds. This discovery indicates higher avialan diversity in the Late Jurassic than previously recognized.
DOI: https://doi.org/10.7554/eLife.43789.001

*For correspondence:
o.rauhut@lrz.uni-muenchen.de

†These authors contributed equally to this work

Competing interests: The authors declare that no competing interests exist.

## Introduction

The so-called 'Solnhofen limestones' of southern Germany have long been known for their exceptionally preserved fossils (see *Arratia et al., 2015*). Historically, most fossils reported from these rocks come from the Altmühltal Formation (sensu *Niebuhr and Pürner, 2014*; see that publication for a clarification of the confusing history of naming the Late Jurassic limestone units of the Franconian Alb), as these were subject to intensive quarrying for several commercial purposes, from construction to lithography, probably since Roman times (*Neumeyer, 2015*). However, fossils have been known from the underlying Torleite and the overlying Mörnsheim formations for a long time (see e.g. *Tischlinger, 2001*), and more recent work in these units has shown that they are, at least at some localities, actually more fossiliferous than the Altmühltal Formation (see e.g. *Viohl and Zapp, 2007*; *Heyng et al., 2011*; *Heyng et al., 2015*; *Albersdörfer and Häckel, 2015*; *Viohl, 2015*).

Arguably the historically most important fossil taxon from the 'Solnhofen limestones' is the famous 'urvogel' *Archaeopteryx*, which has played a crucial role in the debate about the origin of birds (e.g. *Huxley, 1868*; *Heilmann, 1926*; *Ostrom, 1976a*; *Wellnhofer, 2008*; *Wellnhofer, 2009*). This taxon is so far only known from the lower Tithonian of Bavaria, Germany, with the vast majority of specimens (nine out of eleven) being derived from the Altmühltal Formation (*Rauhut et al., 2018*). Only one specimen (the 12th specimen; see comments on numbering of specimens below) comes from the Kimmeridgian/Tithonian boundary in the lowermost part of the Painten Formation (*Rauhut et al., 2018*), and a single, fragmentary skeleton was reported from the Mörnsheim

**eLife digest** The origin of birds and their flight has been heavily debated in the field of evolutionary biology since the late nineteenth century. Birds are the only living descendants of dinosaurs and, for paleontologists, the famous *Archaeopteryx* has played a pivotal role in this discussion. Living during the Jurassic period about 150 million years ago in what is now southern Germany, *Archaeopteryx* is generally accepted as the oldest known flying bird. Yet, with the discovery of other bird-like dinosaurs from the same period, a question has arisen as to whether *Archaeopteryx* is the only flying bird from the Jurassic.

To answer this question, Rauhut et al. carefully examined a fossil of an isolated wing skeleton that was recently discovered in the same region of Germany where *Archaeopteryx* was found. The new specimen shows several characteristics that are otherwise only found in modern birds and not seen in *Archaeopteryx*. As such, this fossil represents a new species and the most bird-like bird discovered from the Jurassic. Rauhut et al. named the species *Alcmonavis poeschli*, after the ancient name for a river that flows near the discovery site, the Greek word for bird, and Roland Pöschl – the collector who found the specimen.

The wing of *Alcmonavis* also shows several features related to the attachment of flight muscles that suggest it was better adapted for flapping flight than *Archaeopteryx*. Together these findings are mostly consistent with the hypothesis that birds first started flying by flapping their wings rather than starting from a gliding stage. However, more detailed studies of the anatomy of primitive birds and their close relatives are needed to further test this hypothesis.
DOI: https://doi.org/10.7554/eLife.43789.002

Formation (*Figure 1*), which overlies the Altmühltal Formation (*Mäuser, 1997*; *Tischlinger, 2009*), and was recently made the type of a new species of this genus, *Archaeopteryx albersdoerferi Kundrát et al., 2019*). In total, the genus *Archaeopteryx* seems to span some 700,000 to one million years, and notable variation between the different specimens might indicate evolutionary changes over this time (*Rauhut et al., 2018*).

For more than 150 years, *Archaeopteryx* was the only Jurassic representative known of the Paraves, the clade of theropod dinosaurs that includes birds (Avialae; see *Gauthier, 1986*) and their closest relatives, dromaeosaurids and troodontids, which are united in a monophyletic clade, Deinonychosauria, in most phylogenetic analyses. However, the discovery of a diverse fauna of paravian theropods from slightly older rocks in China in the last decades (e.g. *Godefroit et al., 2013a*; *Godefroit et al., 2013b*; *Sullivan et al., 2014*; *Xu et al., 2015*; *Lefèvre et al., 2017*), and the identification of the 'Haarlem specimen of *Archaeopteryx*' as a separate taxon, *Ostromia crassipes* (*von Meyer, 1857*), representing a different clade of paravians, the Anchiornithidae (*Foth and Rauhut, 2017*), highlight the complexity of paravian evolution, diversity and distribution in the Late Jurassic, and make a careful re-evaluation of all maniraptoran specimens from the Late Jurassic plattenkalks of southern Germany necessary.

Here we report on a new paravian specimen from the Lower Tithonian Mörnsheim Formation, representing the second theropod specimen from this unit, which overlies the Altmühltal Formation. The new specimen represents the largest avialan theropod yet recorded from the Jurassic and provides further evidence on the forelimb anatomy and the origin of flapping flight in basal avialans.

## Geological and palaeontological context

The Mörnsheim Formation is a unit of the southern German Weißjura Group, a package of mainly calcareous marine sediments that is widely distributed in Bavaria and Baden-Württemberg. In the southern Franconian Alp in Bavaria, the Weißjura Group in the region between Weißenburg and Regensburg is famous for often laminated, very fine-grained limestones of late Kimmeridgian to early Tithonian age, often collectively called 'Solnhofen limestones', and the fossils from these rocks are accordingly referred to as 'Solnhofen fossils'. A large number of often local or regional names have been proposed for the different units that make up the 'Solnhofen limestones', but a recent overview of the lithostratigraphy of the area helped to clarify the nomenclature and correlations of the formations (*Niebuhr and Pürner, 2014*). Thus, the 'Solnhofen limestones' sensu stricto are now

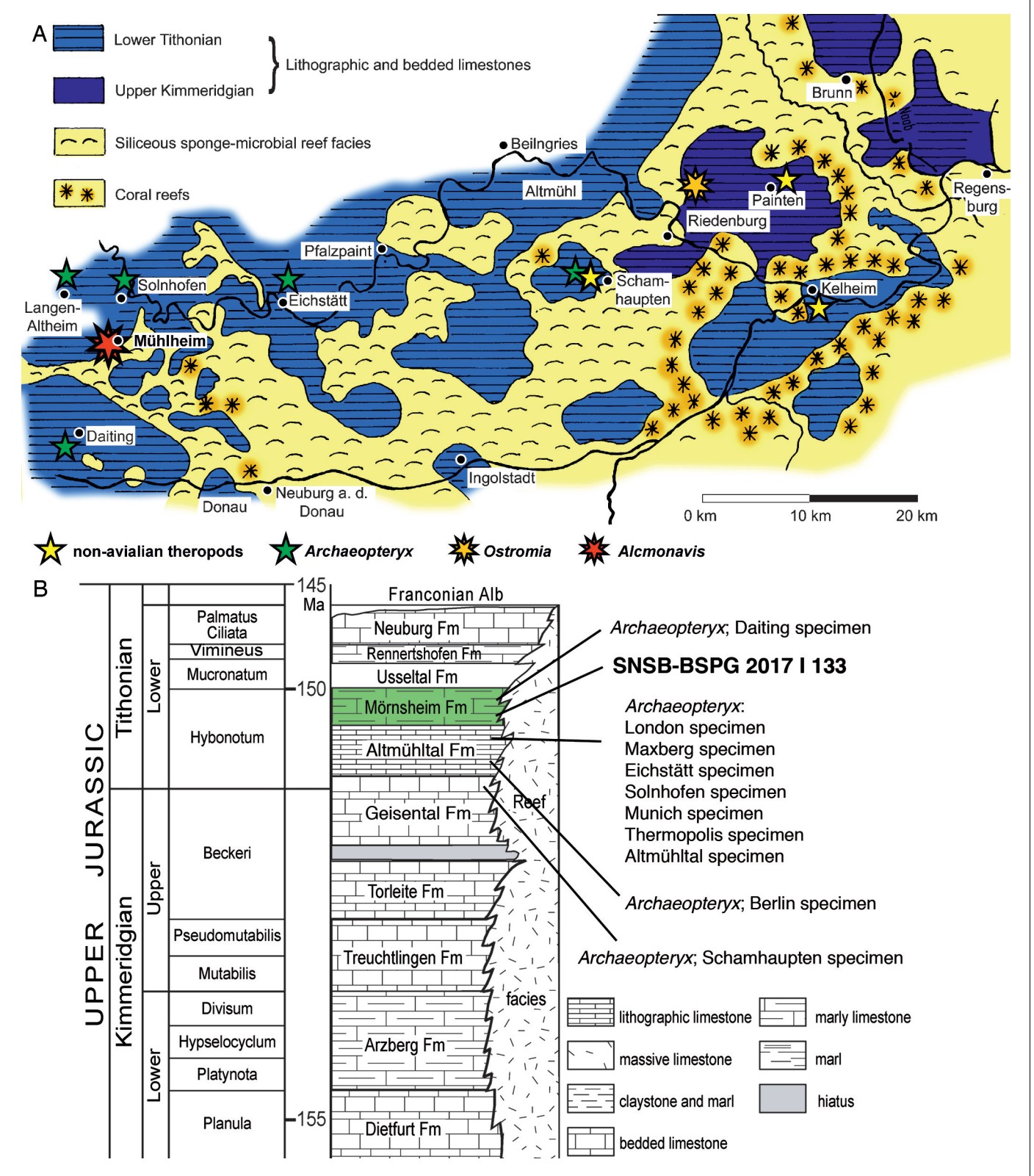

**Figure 1.** Geographic and stratigraphic provenance of the new avialan specimen. (**A**) Map of the southern Franconian Alb with the palaeogeographic settings indicated and showing the localities of theropod specimens from the Solnhofen Archipelago (modified from *Foth and Rauhut, 2017*). (**B**) Stratigraphic position of the new specimen, SNSB-BSPG 2017 I 133, within the 'Solnhofen limestones' and in comparison to known specimens of *Archaeopteryx* (modified from *Rauhut et al., 2012*).

*Figure 1 continued on next page*

Figure 1 continued

DOI: https://doi.org/10.7554/eLife.43789.003

included in the Altmühltal Formation and restricted to the area northwest of Ingolstadt, whereas more eastern occurrences of contemporaneous plattenkalks are included in the Painten Formation. Both of these formations underly the Mörnsheim Formation. Biostratigraphic dating with the help of ammonites has furthermore shown that the Altmühltal Formation spans from the uppermost horizon of the *Beckeri* zone of the latest Kimmeridgian over five ammonite horizons to the *rueppelianus* horizon of the *Hybonotum* zone of the Early Tithonian, and that the lithographic limestones within this formation in the areas of Solnhofen and Eichstätt are not synchronous, but the Eichstätt Member is somewhat older (*Schweigert, 2007*; *Schweigert, 2015*; *Niebuhr and Pürner, 2014*). Furthermore, in more eastern areas, important vertebrate fossils have also been found in the upper part of the Torleite Formation, which underlies the Painten Formation. In the light of this geological and stratigraphic complexity, the traditional habit to talk about all the fossils from these diverse units as the 'fauna of the Solnhofen limestones' has been abandoned in favour of the expression 'fauna of the Solnhofen Archipelago' in recent years, to distinguish the regional palaeoecological setting from the concrete geological unit that the different fossils are derived from (e.g. *Röper, 2005*; *López-Arbarello and Schröder, 2014*; *Rauhut et al., 2017*).

The Mörnsheim Formation has its best outcrops in the areas between Mörnsheim, Solnhofen, Monheim, and Daiting (*Figure 1*). Lithologically, the Mörnsheim Formation differs from the Altmühltal Formation in the considerably higher amount of silicified limestones ('Kieselplattenkalke'), especially in its lower part. Biostratigraphically, this unit represents the uppermost horizon of the *Hybonotum* zone, the *moernsheimensis* horizon, and is thus slightly younger than the Upper Solnhofen Member of the Altmühltal Formation. Fossils have been known for a long time from the Mörnsheim Formation, mainly from the locality of Daiting (*Tischlinger, 2001*), but the fauna of this formation has only partially been explored so far, mainly due to the fact that this formation has not been quarried extensively.

The specimen described here comes from the Schaudiberg, near Mühlheim, close to Mörnsheim (*Figure 1A*). Two quarries are currently exposed at the Schaudiberg, both owned by the Grundstücksgemeinschaft Pöschl/Leonhardt, a public quarry for fossil collectors, and the Old Schöpfel Quarry, which is being systematically excavated for fossils (see *Heyng et al., 2015*). The lower part of the Mörnsheim Formation has a total thickness of approximately 50 m at the Schaudiberg, but only parts of this are exposed in the two quarries. Some 8 m of the lowermost Mörnsheim Formation are exposed in the Old Schöpfel Quarry, with the boundary to the underlying Altmühltal Formation at the base of the section being currently covered. Thus, the currently exposed section starts some 4 m above this boundary with silicified laminated limestones and intercalated thick layers of massive limestones and silicified limestones (*Heyng et al., 2015*). In the higher part of the profile and the visitors quarry, the section becomes more dominated by laminated limestones and intercalations of laminated marly limestones and clays. The new urvogel specimen comes from a thin marly intercalation within the lowermost 2.5 m of the section in the Old Schöpfel Quarry. It was found in 2017 by Roland Pöschl, who leads the systematic excavations in this quarry. The Mörnsheim Formation at the Schaudiberg is very fossiliferous, with the most common fossils being strongly compressed ammonites, and a rich invertebrate and vertebrate fauna is present, but remains largely unstudied so far. In contrast to the underlying Altmühltal Formation, most vertebrate fossils in the Mörnsheim Formation are at least partially disarticulated and often fragmentary. In the Schaudiberg quarries, fishes are represented by chondrichthyans, including well-preserved specimens of *Asteracanthus* (*Pfeil, 2011*), actinopterygians (e.g. *Schröder and López-Arbarello, 2013*), and mainly isolated remains of coelacanths. Tetrapods are represented by unstudied turtles, rhynchocephalians, marine crocodiles, and pterosaurs (*Heyng et al., 2011*; *Heyng et al., 2015*; *Rauhut et al., 2011*; *Rauhut et al., 2012*; *Moser and Rauhut, 2011*; *Rauhut, 2012*), with the only formally studied taxon being the unusual rhynchocephalian *Oenosaurus muehlheimensis Rauhut et al., 2012*, so far. As noted above, the only theropod specimen reported from the Mörnsheim Formation so far is the fragmentary holotype of *Archaeopteryx albersdoerferi*, which comes from the outcrop area of Daiting (*Tischlinger, 2009*; *Kundrát et al., 2019*).

## Comments on the numbering and naming of avialan specimens from the Solnhofen Archipelago

Traditionally, the different specimens found in the Kimmeridgian-Tithonian limestones of Bavaria have been numbered according to the time they were first described as *Archaeopteryx* (see *Wellnhofer, 2008*; *Wellnhofer, 2009*; *Rauhut and Tischlinger, 2015*). Thus, the London specimen is usually called the first skeletal specimen, the Berlin specimen the second and so on. Accordingly, more recently, *Foth et al., 2014* described a new specimen that is hitherto simply known as the 11[th] specimen, and *Rauhut et al., 2018* described the 12[th] specimen. However, the story becomes more complicated if one accepts that some specimens might actually not belong to the genus *Archaeopteryx*, such as the Solnhofen specimen, which was argued to represent a distinct genus, *Wellnhoferia* (*Elzanowski, 2001*; *Elzanowski, 2002*; though see *Mayr et al., 2007*, and *Wellnhofer, 2008*, *Wellnhofer, 2009*), or the Haarlem specimen, which was recently referred to a distinct genus of anchiornithids, *Ostromia* (*Foth and Rauhut, 2017*). Thus, in this case, theoretically, the numbering of *Archaeopteryx* specimens would need to be revised, with the 11[th] specimen becoming the 10[th] specimen and so forth. However, we propose to retain the original numbering of specimens, even if one accepts the different generic assignments, in order to avoid confusion between the recent and older literature. Given the gradual assembly of the avialan body plan (*Brusatte et al., 2014*; *Cau et al., 2015*) and the general similarity of the basalmost members of this clade, it might be justified to simply talk about 'urvogel specimens' instead of using the generic name *Archaeopteryx*, to thus accommodate the taxonomic uncertainty. Accordingly, the specimen described here should be regarded as the 13[th] urvogel specimen from the Solnhofen Archipelago.

Apart from the numbers, most specimens also have informal, but widely used names, usually based on their repository ('London specimen', 'Berlin specimen', 'Eichstätt specimen', etc.). In contrast to many other fossil taxa, it is these names, often in combination with the numbering outlined above, rather than specimen numbers that are usually used to identify the different specimens of *Archaeopteryx* and possible other basal avialans from the Kimmeridgian/Tithonian of Bavaria, even in the technical scientific literature (e.g. *Elzanowski, 2002*; *Mayr et al., 2007*; *Wellnhofer, 2008*; *Wellnhofer, 2009*; *Rauhut et al., 2018*). Unfortunately, however, the most recently described specimens have so far only been identified by their numbers, and no names have been proposed. Thus, in order to facilitate communication about the specimens, we here propose the following names for specimens 11 to 13:

11[th] specimen: This specimen represents an almost complete postcranial skeleton and parts of the skull of *Archaeopteryx*, preserved in articulation and with exceptionally detailed feather impressions (*Foth et al., 2014*). The specimen most probably comes from Upper Eichstätt Member of the Altmühltal Formation of the Schernfeld/Blumenberg area, close to Eichstätt (M. Röper, pers. com. to OR, 05.2018) and is currently on exhibition at the Museum Solnhofen. However, as neither the exact provenance nor the final repository of this specimen are certain by now, we propose to refer to it as the 'Altmühl specimen', referring to both its general area of provenance close to the Altmühl river and the geological unit it is derived from.

12[th] specimen: This specimen is an almost complete, although partially poorly preserved, largely articulated skeleton of *Archaeopteryx* (*Rauhut et al., 2018*). It comes from the lowermost parts of the Painten Formation of the Gerstner Quarry in Schamhaupten and is currently on exhibition at the Dinosaurierpark Altmühltal in Denkendorf. Due to its provenance, we propose to name this specimen the 'Schamhaupten specimen'.

13[th] specimen: This specimen is described here. It comprises an associated right wing of a large basal avialan from the Mörnsheim Formation of Mühlheim, close to Mörnsheim, Bavaria. The specimen belongs to the collections of the Bayerische Staatssammlung für Paläontologie und Geologie in Munich. As there already is a 'Munich specimen', we propose to refer to this specimen as the 'Mühlheim specimen'.

### Institutional abbreviations

AMNH, American Museum of Natural History, New York, USA; IGM, Institute of Geology, Ulan Bataar, Mongolia; IVPP, Institute of Vertebrate Paleontology and Paleoanthropology, Beijing, China; JME, Jura-Museum Eichstätt, Germany; JZT, Jizantang Paleontological Museum, Chaoyang City, China; MCF, Museo Carmen Funes, Plaza Hiuncul, Argentina; SNSB-BSPG, Staatliche

naturwissenschaftliche Sammlungen Bayerns, Bayerische Staatssammlung für Paläontologie und Geologie, Munich, Germany.

## Description

### Preservation

The new specimen, SNSB-BSPG 2017 I 133, consists of the partially disarticulated, but associated skeleton of the right wing of an avialan theropod (*Figure 2*). The humerus is rotated and slightly displaced from the remains of the arm, and radius and ulna are disarticulated, but preserved in close proximity. The manus is disarticulated from the radius and ulna and the metacarpus overlies the distal ulna. The digits of the hand are mainly preserved in articulation, with only phalanx I-1 forming an unnatural angle with its respective metacarpal, and the ungual of digit I being displaced, lying at the distal end of the ulna.

The head of the humerus is not preserved, as it lay in a mud-filled crack within the rock (U. Leonhardt, pers. com. to OR, 10.2017) and the proximal shaft of the radius is partially reconstructed. The bone preservation is generally rather poor, with all longbones being compressed, collapsing the shafts, and fractured, and the entepicondyle of the humerus is largely lost, as is the distal end of metacarpal I. However, the phosphatized remains of the original keratinous sheath of the unguals

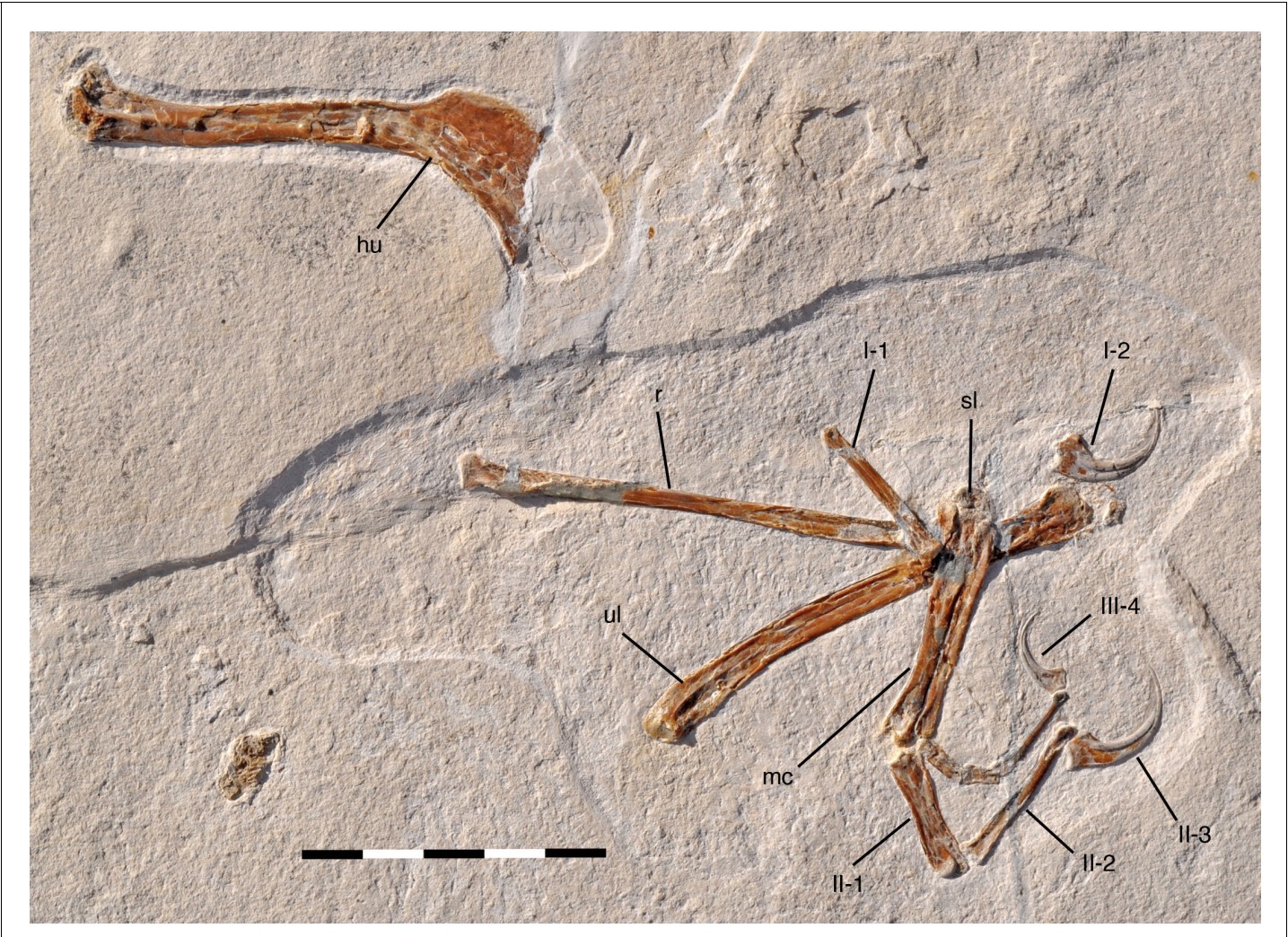

**Figure 2.** Overview photograph of holotype specimen of *Alcmonavis poeschli* gen. et sp. nov., SNSB-BSPG 2017 I 133. Abbreviations: hu, humerus; mc, metacarpus; r, radius; sl, semilunate carpal; ul, ulna; Roman numerals indicate digits and Arabic numerals indicate phalanges of digits. Scale bar is 5 cm.
DOI: https://doi.org/10.7554/eLife.43789.004

are rather nicely preserved, as it is often the case in theropod specimens from the Late Jurassic plattenkalks of southern Germany.

## Humerus

The right humerus is exposed in anteromedial view (*Figure 3*). As noted above, the bone is missing its proximal end, so its exact length cannot be established (the apparent outline of the bone in the sediment is an artefact of preparation, as the outline has been sculpted in the matrix used to fill the crack). However, the medial edge preserves the distal end of the internal tuberosity, indicating that only a small portion of the proximal end is missing, so that the complete length can be estimated to be approximately 90 mm (±2.5 mm; length as preserved: 81.4 mm). As in all maniraptoran theropods (*Rauhut, 2003*), the internal tuberosity was obviously proximodistally elongate and rectangular in outline, with approximately its distal third being preserved. Distally, it fades into the relatively sharp medial edge of the shaft in an oblique step. On the anterolateral side of the humerus, the distal half of the large, rectangular deltopectoral crest is preserved. Whereas the anterolateral edge of the proximal part of the crest seems to be rather sharp-edged, the distalmost c. 11 mm show an elongate oval facet for the insertion of the *m. pectoralis* (*Figure 3*). This facet is inclined anteromedially and reaches a maximal transverse width of 2.2 mm. A similar facet is also developed in *Sapeornis* (*Provini et al., 2009*; pers. obs. by CF on specimen JZT-DB 0047), *Jeholornis* (*Lefèvre et al., 2014*), *Jixiangornis* (*Chiappe and Meng, 2016*), *Confuciusornis* (e.g. JME 1997/1; *Chiappe et al., 1999*), but it is absent in *Archaeopteryx*, anchiornithids and non-avian theropods (see discussion). Distal to this insertion, the deltopectoral crest fades into the anterolateral side of the shaft in an oblique, slightly concave step. As in all specimens of *Archaeopteryx*, other basal avialans like *Confuciusornis* (*Chiappe et al., 1999*), and, to a lesser degree, in non-avialan maniraptorans (e.g. *Ostrom, 1969*; *Clark et al., 1999*; *Pei et al., 2014*), the proximal part of the humerus that houses the deltopectoral crest is angled posteriorly in respect to the central shaft, with its posteromedial edge (excluding the internal tuberosity) forming an angle of approximately 38° towards the long axis of the shaft. This angle is larger than in non-avialan Paraves and *Anchiornis* (e.g., *Ostrom, 1969*; *Burnham, 2004*; *Pei et al., 2014*; *Pei et al., 2017*). In *Archaeopteryx*, the angle varies between 30° (Daiting specimen) and 33° (Maxberg and Ottman and Steil specimens), while it is similarly increased in more derived avialans (see discussion).

The length of the humerus distal to the deltopectoral crest is 64.8 mm as measured from the distalmost edge of the apex of the crest and c. 60 mm from the point where the distal end of the crest fades into the shaft. The humeral shaft distal to the deltopectoral crest is very slightly sigmoidal, with the proximal c. 30 mm being slightly convex anteriorly and concave posteriorly, whereas the distal end is slightly flexed anteriorly. Although the compaction of the shaft makes secure interpretation difficult, it seems that it was wider transversely than anteroposteriorly over its entire length. The minimal width of the shaft is c. 6.5 mm (all measurements should be seen with caution due to the compaction of the bones). The distal end gradually expands transversely to approximately 200% of the minimal shaft width (as far as this can be established; 13 mm as preserved) and the distal condyles expand anteriorly. A large, deep, triangular groove is present proximal to and between the condyles on the anterior side of the humerus (*Figure 3*). Although this groove might be somewhat exaggerated by the compaction of the bone, it nonetheless seems to be a genuine character of the bone. The medial rim of the groove is formed by a broad longitudinal ridge that is slightly offset laterally from the medial margin of the distal humerus and fades into the shaft some 15 mm proximal to the distal end. This groove and ridge have not been described for *Archaeopteryx* so far (due to the fact that the anterior side of the distal humerus is not exposed in any of the specimens referred to this genus), but they are also present in the humeri of the dromaeosaurids *Deinonychus* (*Ostrom, 1969*), and *Bambiraptor* (*Burnham, 2004*), in *Balaur* (*Brusatte et al., 2013*) and the basal avialans *Confuciusornis* (*Chiappe et al., 1999*) and *Sapeornis* (*Provini et al., 2009*). In modern birds, a depression in a similar position serves as the attachment of the brachialis muscle, the *fossa musculi brachialis* (*Baumel and Witmer, 1993*), and this might be a likely identification in these paravian theropods as well. The entepicondyle (ulnar condyle or condyles dorsalis) is largely missing, but its impression in the rock indicates that it was anteriorly expanded and posteromedially rounded. The ectepicondyle (radial condyle or condyles ventralis) is considerably expanded anteriorly and has a

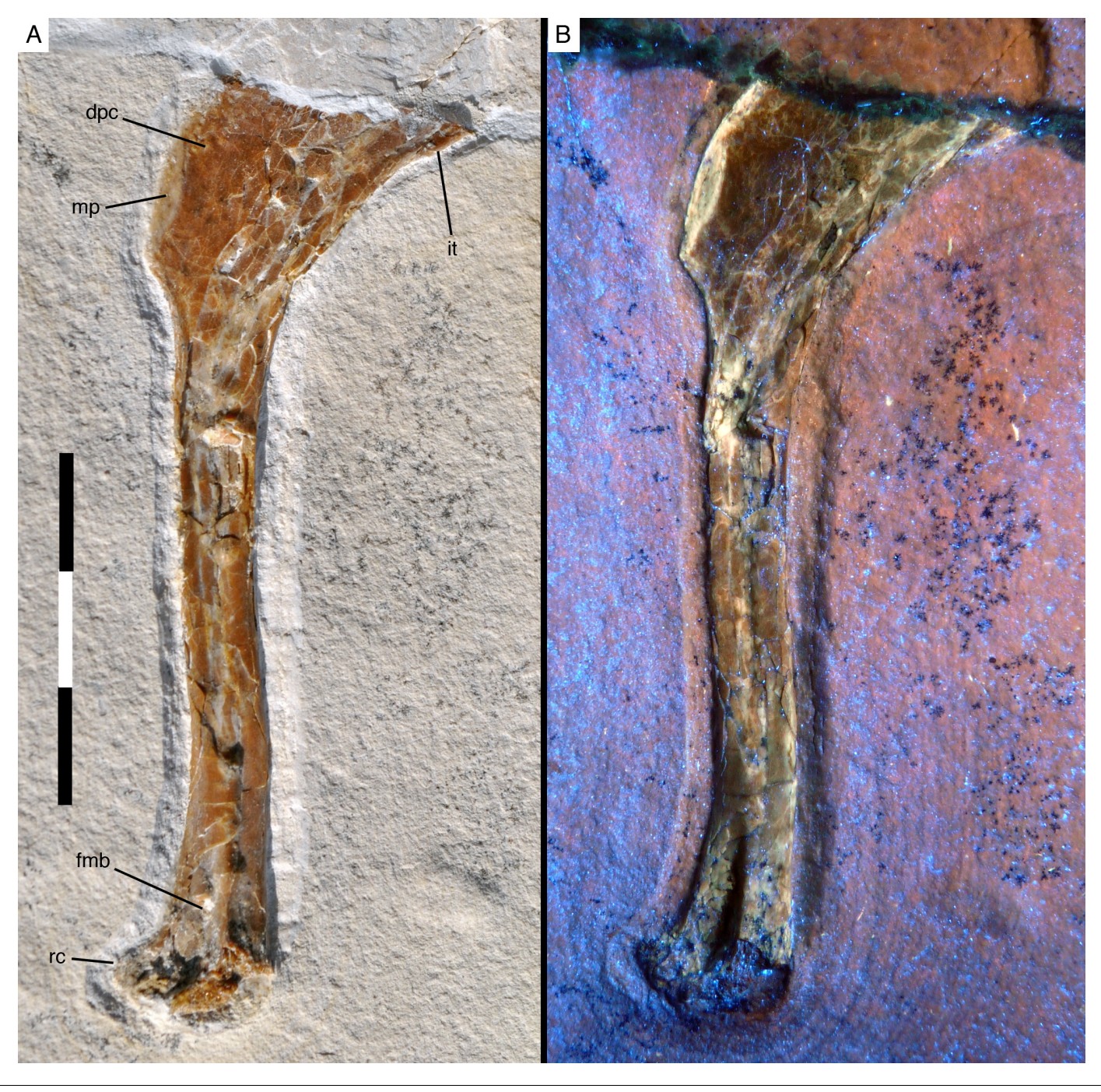

**Figure 3.** Right humerus of *Alcmonavis poeschli* in (A) normal and (B) ultraviolet light. Abbreviations: dpc, deltopectoral crest; fmb, *fossa musculus brachialis*; it, internal tuberosity; mp, attachment facet for *m. pectoralis*; rc, radial condyle. Scale bar is 3 cm.
DOI: https://doi.org/10.7554/eLife.43789.005

well-developed, anterodistally facing, transversely concave roller joint, similar to the condition in *Deinonychus* (*Ostrom, 1969*).

## Ulna
The ulna is a slender element (*Figure 4*) and, with a length of 82 mm, slightly shorter than the estimated length of the humerus, as in *Archaeopteryx* (*Wellnhofer, 2008*; *Wellnhofer, 2009*;

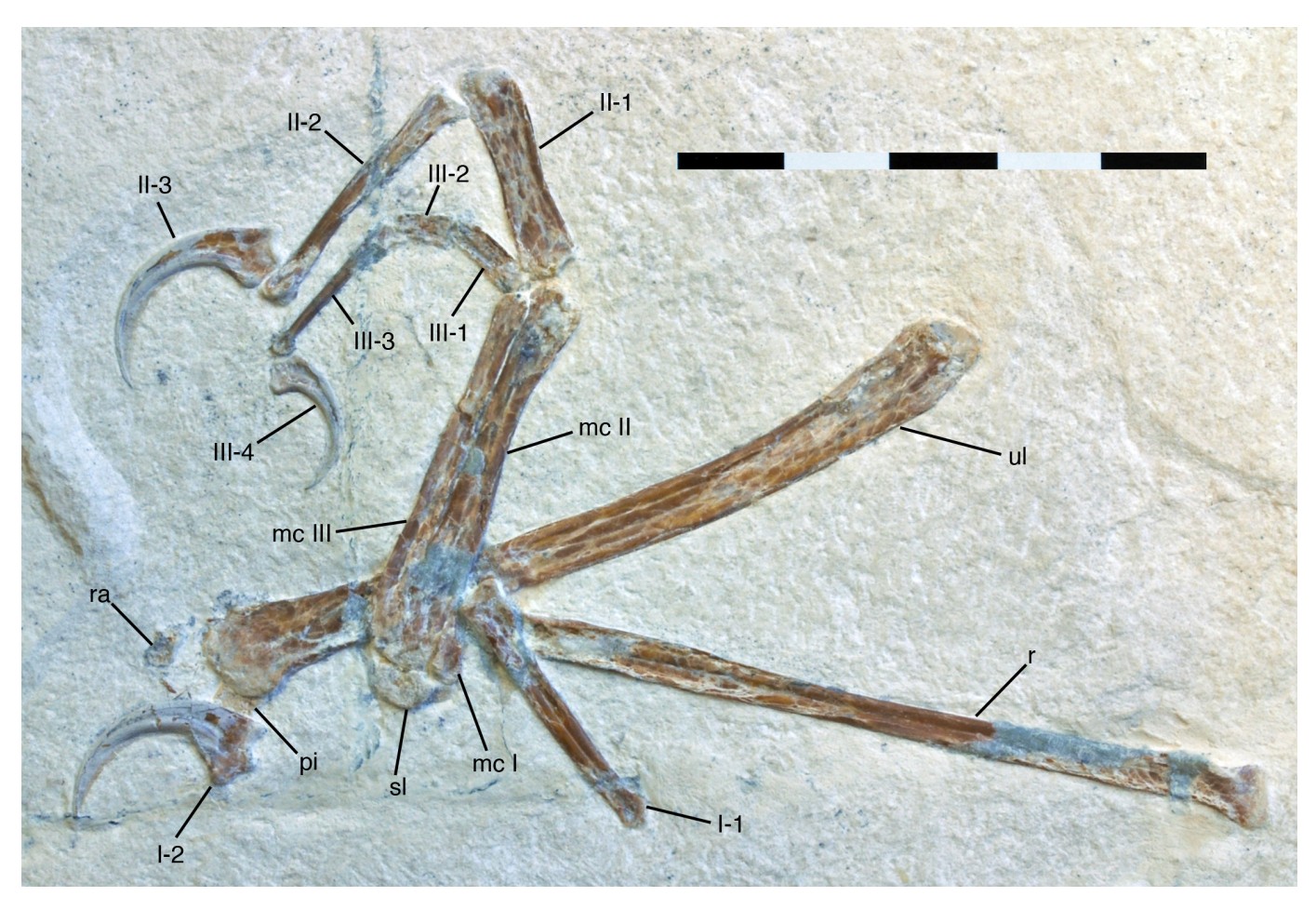

**Figure 4.** Antebrachium and manus of *Alcmonavis poeschli*. Abbreviations as in *Figure 1*, and: pi, pisiforme; ra, radiale. Scale bar is 5 cm.
DOI: https://doi.org/10.7554/eLife.43789.006

*Foth et al., 2014*; *Rauhut et al., 2018*), non-avialan maniraptorans, *Confuciusornis* (e.g. *Chiappe et al., 1999*), and *Apsaravis* (*Clarke and Norell, 2002*), whereas both elements are of subequal length in *Sapeornis* (e.g. *Zhou and Zhang, 2003a*; *Provini et al., 2009*), and the ulna is longer than the humerus in many more derived avialans (e.g. *Hu et al., 2014*; *Wang et al., 2016*). It is exposed in anteromedial view. As in many other maniraptoran theropods (*Gauthier, 1986*), the proximal half of the ulna is slightly bowed, being convex posteriorly. The proximal end of the ulna is only slightly expanded from the shaft, and the olecranon process is poorly developed (*Figure 5*), as in most maniraptorans, including birds (*Rauhut, 2003*). The proximal articular surface shows a small, almost round, proximally and very slightly anteriorly facing concavity anteriorly, and a smaller convexity posteriorly. Thus, in contrast to *Deinonychus* (*Ostrom, 1969*), other dromaeosaurids (*Norell and Makovicky, 1999*; *Hwang et al., 2002*; *Burnham, 2004*), and *Balaur* (*Brusatte et al., 2013*), but similar to *Jeholornis* (*Lefèvre et al., 2014*; *Chiappe and Meng, 2016*) the proximal end is oval rather than triangular in outline in proximal view.

Below the proximal articular surface, a small, but sharply defined longitudinal ridge is present on the anteromedial side of the ulna (*Figure 5*). This ridge extends from the anteromedial edge of the proximal articular surface, is slightly offset from the medial side of the bone and fades into the shaft some 9 mm below the articular surface. A very similar ridge is present in the 12th specimen of *Archaeopteryx* (*Rauhut et al., 2018*: fig. 22), *Confuciusornis* (JM 1997/1), *Ichthyornis* (*Clarke, 2004*: fig. 52C) and many modern birds, where it marks the margin of the *impressio brachialis* (*Baumel and Witmer, 1993*), but such a ridge is absent in non-avialan theropods. Laterally, a small lateral tubercle

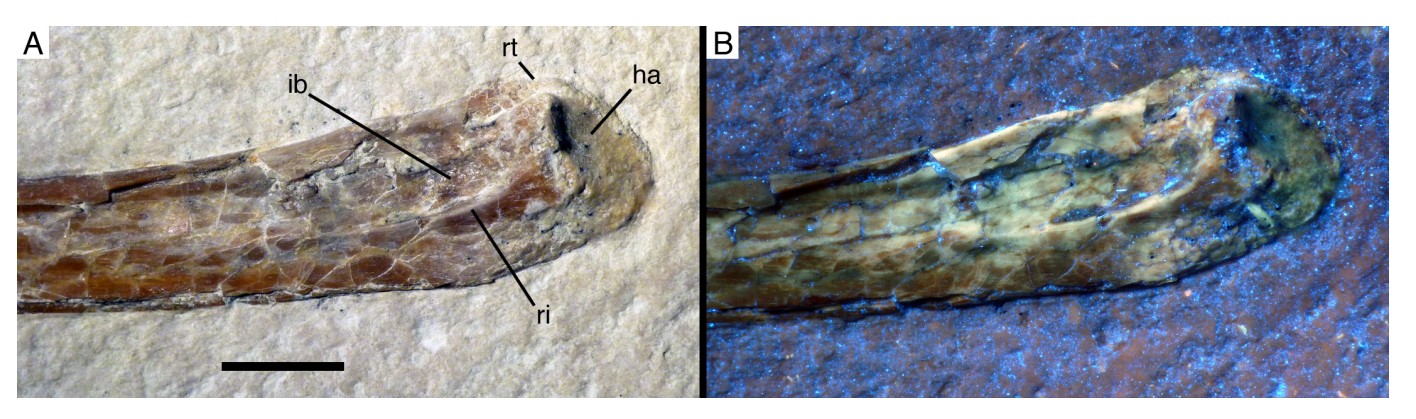

**Figure 5.** Proximal end of ulna of *Alcmonavis poeschli* in (**A**) normal and (**B**) ultraviolet light. Abbreviations: ha, humeral articulation; ib, *impressio brachialis*; ri, ridge bordering the *impressio brachialis*; rt, radial tubercle (*cotylus dorsalis*). Scale bar is 5 mm.
DOI: https://doi.org/10.7554/eLife.43789.007

is present proximally, defining the posterior border of the radial fossa and rapidly disappearing into the ulnar shaft distally. The tubercle is offset from the ulnar articular surface by a ridge, but its proximal surface is convex and does not show an additional articular surface, as it is the case in the cotyla dorsalis of modern birds. A pronounced tubercle for the insertion of the biceps muscle cannot be identified, but the bone is strongly crushed in the area where this tubercle would be expected.

The shaft is slender and narrows gradually, but only slightly from approximately 7 mm at the proximal end to approximately 5 mm at its mid-length. From there, it expands again towards the distal end, which seems to be anteroposteriorly flattened and expanded transversely (*Figure 6*), as in for example the dromaeosaurid *Bambiraptor* (*Burnham, 2004*), the enantiornithine *Rapaxavis* (*O'Connor et al., 2011*) or the ornithuromorphs *Archaeorhynchus* (*Zhou et al., 2013*) or *Gansus* (*Wang et al., 2016*). Whereas the lateral expansion is more gradual, the medial side has an abrupt, rounded expansion that starts some 6 mm proximal to the distal end forming a dorsal ulnar condyle (in *Rapaxavis* and *Archaeorhynchus* the distal end of the ulna has a gradual medial expansion and an abrupt lateral expansion; *O'Connor et al., 2011*; *Zhou et al., 2013*). In contrast, the distal ulna of

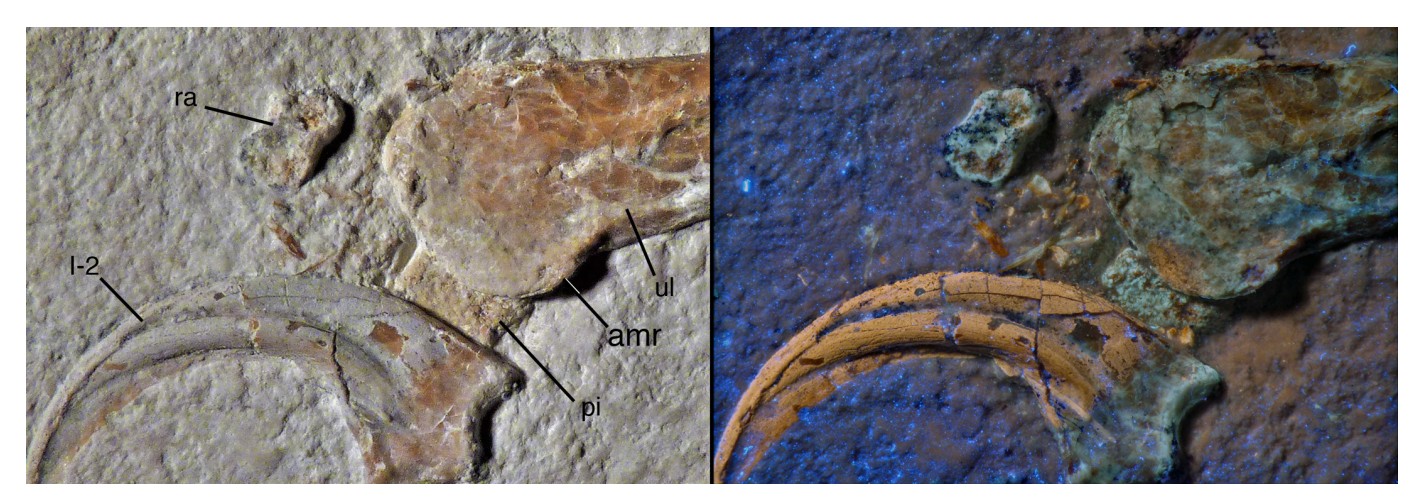

**Figure 6.** Distal end of ulna and proximal carpals of *Alcmonavis poeschli* in (**A**) normal and (**B**) ultraviolet light. Abbreviations as in *Figures 2* and *4* and: amr, anteromedial ridge. Scale bar is 5 mm.
DOI: https://doi.org/10.7554/eLife.43789.008

*Archaeopteryx* seems to be less abruptly and more symmetrically expanded. The proximal end of the condylus dorsalis forms a small, anteromedially directed ridge, whereas the expansion flexes gradually into the distal side distally. The general shape of the distal ulna resembles that of the Late Cretaceous ornithurine *Ichthyornis* (*Clarke, 2004*: Figure 52C) and the posterior side of the ulna in modern birds, but it differs from the latter in the absence of a *tuberculum carpale* (*Baumel and Witmer, 1993*). The maximal distal expansion is 9 mm as preserved. The distal articular surface is gently rounded transversely, flexing higher proximally on the medial than on the lateral side (*Figure 6*). However, it is rather poorly preserved, so nothing can be said about its details.

## Radius

As in all theropods, the radius is more slender than the ulna (*Figure 4*), its minimal shaft width being 3 mm. The bone is straight, and its length cannot be established precisely, as the distal end is overlain by the metacarpus, but it is approximately 80–82 mm. The radius seems to be mainly exposed in medial view.

Although the shaft generally widens gently proximally, the proximal end is rather rapidly expanded to an anteroposterior width of c. 6 mm, with this abrupt expansion starting some 4 mm below the proximal end (*Figure 7A*). In contrast to the situation in *Bambiraptor* (*Burnham, 2004*), *Anchiornis* (*Pei et al., 2017*), *Balaur* (*Brusatte et al., 2013*), and *Confuciusornis* (*Chiappe et al., 1999*), but similar to *Deinonychus* (*Ostrom, 1969*), the anterior and posterior sides are almost equally expanded. The proximal articular end extends slightly more proximally posteriorly, where the articular surface is slightly convex and gradually curves into the posterior side of the shaft, whereas the anterior two thirds of the articular surface are gently concave.

An unusual feature of the proximal radius is a raised, medially directed crest on the anteromedial side of the shaft *Figure 7A,B*). This crest expands abruptly medially some 3.5 mm distal to the proximal articular surface and extends for at least 9 mm distally, before it obliquely disappears into the shaft (although the crest is damaged in parts, enough is preserved to establish its approximately shape and extent; *Figure 7B*). At its highest section, the crest extends at least 1.5 mm from the shaft medially. In position and general appearance, this crest corresponds to the *tuberculum bicipitale radii* of birds (*Baumel and Witmer, 1993*), and we thus interpret it as the insertion of the *M. biceps brachii*. In basal paravian theropods, a similar crest on the proximal radius has so far only been

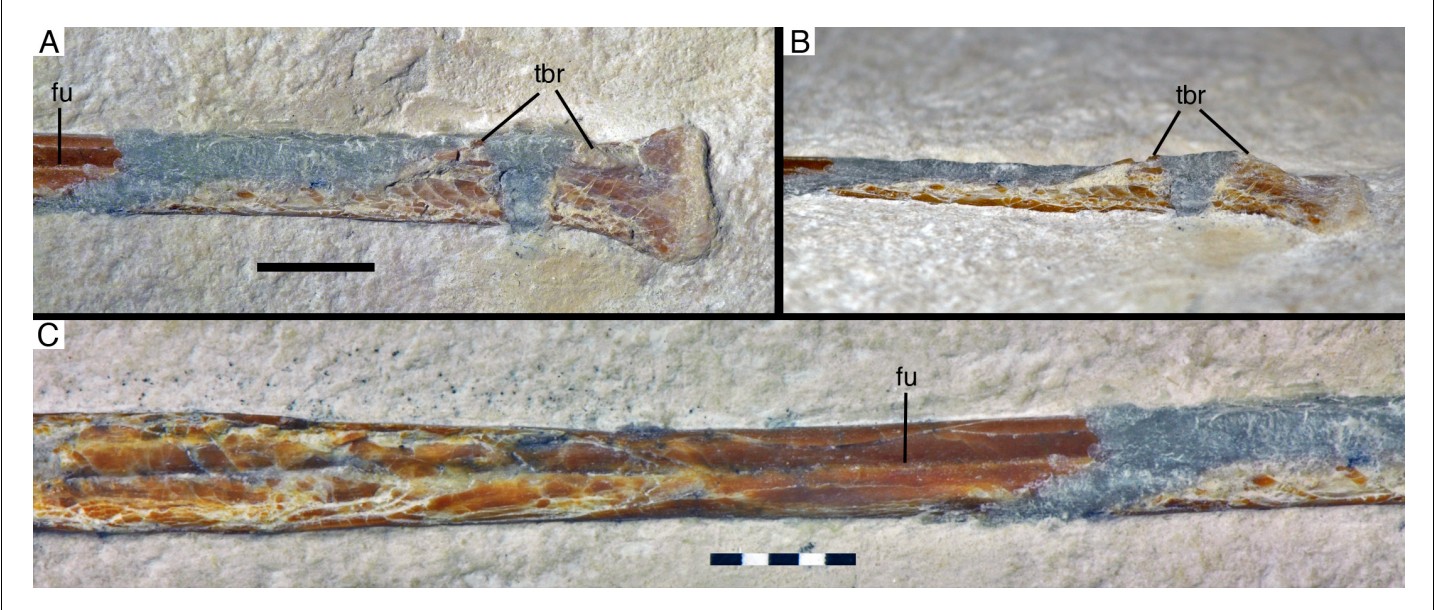

**Figure 7.** Radius of *Alcmonavis poeschli*. (**A**) Proximal end as exposed in medial view. (**B**) Proximal end in oblique posteromedial view. (**C**) mid-shaft in medial view. Abbreviations: fu, longitudinal furrow; tbr, *tuberculum bicipitale radii*. Scale bars are 5 mm.
DOI: https://doi.org/10.7554/eLife.43789.009

described in Bambiraptor (*Burnham, 2004*), but it is absent in *Deinonychus* (*Ostrom, 1969*), *Pyroraptor* (*Allain and Taquet, 2000*), *Anchiornis* (*Pei et al., 2017*), and *Balaur* (*Brusatte et al., 2013*). In *Confuciusornis* (*Chiappe et al., 1999*) and at least some specimens of *Archaeopteryx* (pers. obs.), a small *tuberculum bicipitale radii* is present, but it is triangular in shape and less pronounced than in SNSB- BSPG 2017 I 133 (see below).

The midsection of the radial shaft is more or less intact, showing a longitudinal groove running along the medial side of the shaft (*Figure 7C*). Such a groove is absent in *Archaeopteryx*, but has been described for some species of Enantiornithes (e.g., *Chiappe and Walker, 2002*; *Sanz et al., 2002*; *Hu et al., 2015*). A similar groove was also described for *Jeholornis*, but interpreted as the result of crushing (*Lefèvre et al., 2014*). However, as such groove is also present in other specimens of *Jeholornis*, including the holotype (*Chiappe and Meng, 2016*): 32, 36), this structure might be authentic. The shaft has its minimal width at about mid-length, from where it again gradually expands distally to a width of 4 mm at the point where the radius is overlain by the metacarpus. Nothing can be said about the morphology of the distal end of the bone, as it suffered severe damage.

## Carpus

The carpus is represented by the semilunate carpal that is preserved in articulation with the metacarpus (*Figure 4*), two small bones preserved below the distal end of the ulna (*Figures 6* and *8*), and a very small element preserved in articulation with the proximal end of metacarpal III (*Figures 8* and *9*). In agreement with *Botelho et al., 2014*, the small element articulated with metacarpal III is interpreted as distal carpal 3. The other two carpals are poorly preserved, but probably represent the radiale, preserved a small distance away below the lateral side of the distal ulna, and the pisiforme (=ulnare; see *Botelho et al., 2014*), which is preserved directly below the medial side of the ulna and is partially covered by the first manual ungual.

Both proximal carpals have a rather poorly defined, granular bone surface, probably indicating incomplete ossification (*Figure 6*). The radiale is a small element, maximally 4.3 mm wide and 2.8 mm deep. It is roughly trapezoidal in outline, has rounded edges, is slightly waisted, and tapers towards its medial(?) side. The exposed (distal?) surface is subdivided by an oblique ridge into two slightly angled facets, a larger, concave lateral(?) facet, presumably for the articulation with the trochlea of the semilunate carpal, and a smaller, triangular lateral facet towards the tapering edge, which might represent the attachment of a propatagial tendon (see *Wang et al., 2017*, for the presence of propatagia in maniraptoran theropods).

The pisiforme is slightly larger, being approximately 4.8 mm wide transversely (*Figure 6*). It is poorly preserved and partially overlapped by the distal ulna and the manual ungual I, so nothing can be said about its exact shape. However, the bone seems to have a triangular, posteroproximally tapering process at its posteroproximal corner, possibly for the attachment of the *m. flexor carpi ulnaris* (see *Vazquez, 1994*).

The semilunate carpal is considerably larger (*Figures 8* and *9*), being 6.4 mm wide transversely, and maximally 3.7 mm deep proximodistally on the palmar side. It is preserved in articulation with metacarpal II, but is separated from the latter by a clear sutural line and even slightly detached from metacarpal I on the palmar side, and was thus not fused into a carpometacarpus, similar to the situation found in non-avian Pennaraptora (*Botelho et al., 2014*), and other basal avialans like *Archaeopteryx* (*Wellnhofer, 2009*), *Sapeornis* (*Zhou and Zhang, 2003a*; *Botelho et al., 2014*) and *Confuciusornis* (*Chiappe et al., 1999*). The proximal articular surface is strongly convex transversely, with the medial side being more strongly convex than the lateral, so that the proximalmost point of the carpal is slightly placed medial to its mid-width (*Figure 9*). The distal side of the carpal is subdivided into a larger, gently concave lateral facet for the contact with metacarpal II and a much smaller, more strongly concave facet for metacarpal I. The plantar side of the semilunate carpal is exposed below the medial side of metacarpal I. It extends further medially than the palmar side, bordering the entire plantar medial expansion of the proximal articular surface of metacarpal I, as in *Archaeopteryx* and more basal paravians. Assuming a similar lateral expansion as the palmar side of the carpal, the plantar side is approximately 9.5 mm wide, but only 2.3 mm deep proximodistally at the point where the exposed section disappears below the palmar trochlea. The palmar ridge of the articular trochlea of the semilunate carpal is thus shorter and more strongly convex than the plantar

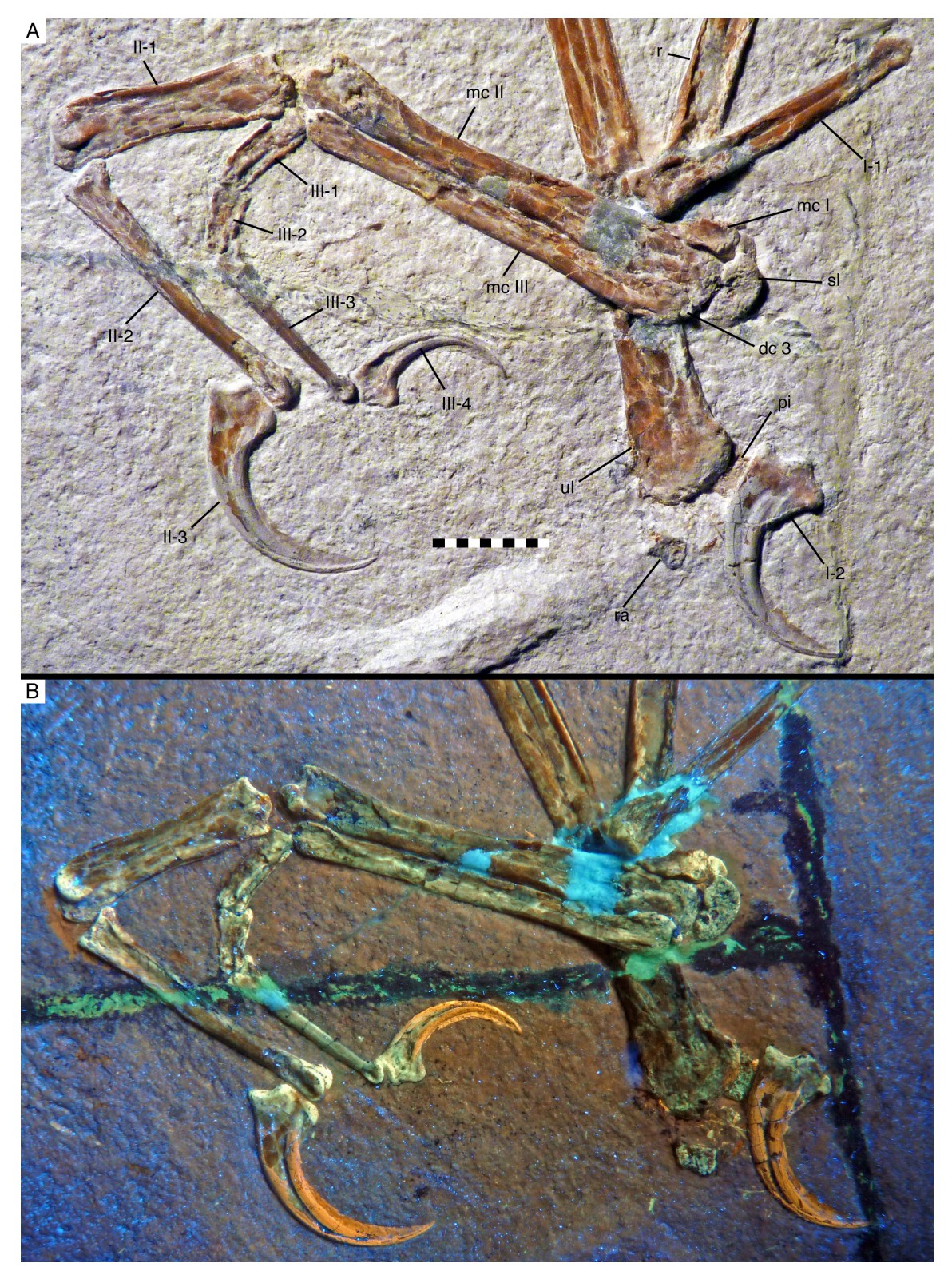

**Figure 8.** Manus of *Alcmonavis poeschli* in (**A**) normal and (**B**) ultraviolet light. Abbreviations as in *Figures 2* and *4*, and: dc, distal carpal. Scale bar is 10 mm.

DOI: https://doi.org/10.7554/eLife.43789.010

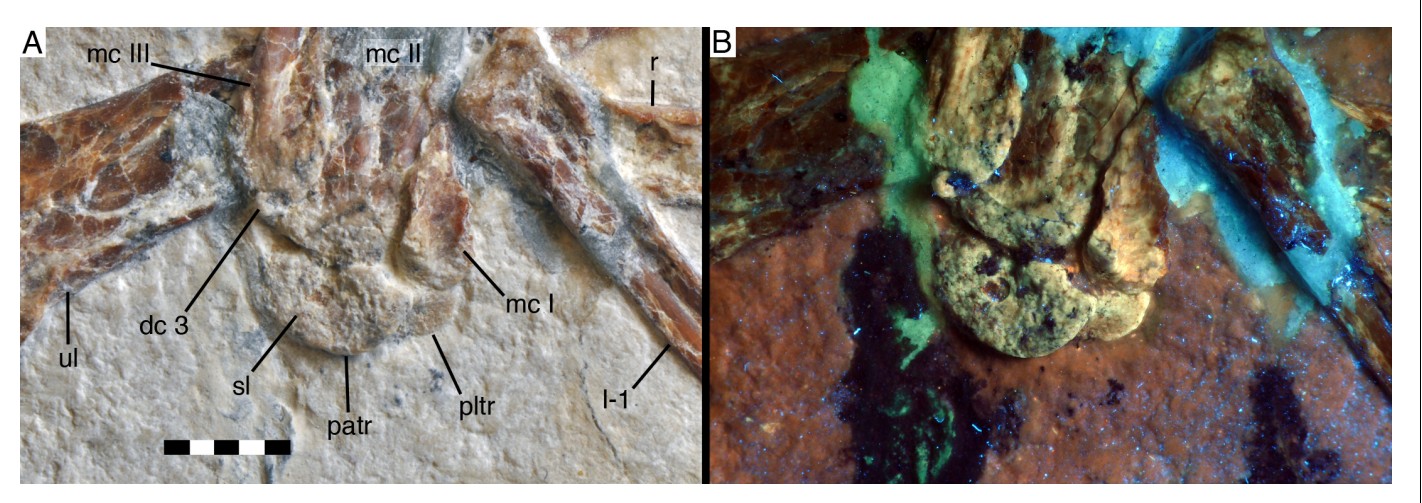

**Figure 9.** Distal carpals and bases of metacarpals of *Alcmonavis poeschli* in (**A**) normal and (**B**) ultraviolet light. Abbreviations as in *Figures 2*, *4* and *8*, and: patr, palmar trochlear ridge; pltr, plantar trochlear ridge. Scale bar is 5 mm.
DOI: https://doi.org/10.7554/eLife.43789.011

ridge of the trochlea (*Figure 9*), as in other paravian theropods, such as *Deinonychus* (*Ostrom, 1969*), *Bambiraptor* (*Burnham, 2004*) and other taxa (see *Xu et al., 2014*). Plantar and palmar trochlear ridges seem to be separated by a deep transverse groove on the proximal articular surface. The lateral edge of the semilunate carpal overlaps half of the area of the proximal end of distal carpal three and metacarpal III, although the latter are offset distally from the carpus.

Distal carpal three is a very small, round nubbin of bone some 1 mm wide transversely and slightly less deep proximodistally, which is attached to the proximal end of metacarpal III.

## Metacarpus

As in all maniraptorans, the metacarpus consists of three elements (*Figure 8*), which are here identified as metacarpals I-III, in agreement with the vast majority of works on theropods (but see *Xu et al., 2009*, for an alternative interpretation). The metacarpus is preserved in articulation and exposed in palmar view. Metacarpal II is notably more robust than either metacarpal I or III (*Figure 8*), with the width of its proximal shaft (c. 5 mm) being approximately twice that of metacarpal III (2.5 mm) and more than 185% of the width of the proximal articular surface of metacarpal I (2.7 mm). This differs from the more equal widths of metacarpal I and II in *Archaeopteryx* (e.g. *Wellnhofer, 1974*; *Wellnhofer, 2008*; *Mayr et al., 2007*), but also the less marked difference in other basal avialans, such as *Sapeornis* (*Zhou and Zhang, 2003a*) and *Confuciusornis* (*Chiappe et al., 1999*). In contrast, these proportions resemble the condition found in *Jeholornis* (*Zhou and Zhang, 2002*; *Lefèvre et al., 2014*).

Metacarpal I is unfortunately incompletely preserved, so that its exact length and maximum width cannot be established (*Figures 8* and *9*). However, the position of the proximal end of phalanx I-1 might indicate the distal end of this metacarpal, in which case it would have been little more than 7 mm long, or about 17–17.5% of the length of metacarpal II and thus relatively much shorter than in *Archaeopteryx* (25–31%). The proximal articular surface of metacarpal I is 3 mm wide transversely and semilunate in outline, with a plantar-palmarly expanded lateral side and a transversely expanded plantar side (*Figure 9*). As the plantar side is embedded in the sediment, it cannot be said whether a plantar process is present laterally, as it is the case in the dromaeosaurids *Deinonychus* (*Ostrom, 1969*) and *Bambiraptor* (*Burnham, 2004*). The mediopalmar side of the proximal end shows a notable depression, distal to which the medial margin seems to have been somewhat expanded, but is broken off. The lateral side of metacarpal I is closely appressed to metacarpal II over its entire preserved length, and the proximal part of the palmar side forms a small lateral flange

that overlaps the mediopalmar edge of metacarpal II (*Figure 9*). The distal condyles are broken off, so nothing can be said about the structure of the distal articulation.

Metacarpal II is massive, apparently subrectangular in cross-section and largely straight, with only the distal articular end being very slightly flexed (*Figure 8*). The exact width of the proximal end is difficult to measure, as its palmar side is partially overlapped by flanges of metacarpals I and III, but it is approximately 5.8 mm. From there, the bone gradually tapers over its proximal two thirds to a minimal width of 3.2 mm some 25 mm from the proximal end, distal to which it expands again slightly. The distal end is slightly flexed medially and exposed somewhat obliquely, and has a maximal width of approximately 4.7 mm. The palmar surface of the distal end is collapsed, but the distal articulation was obviously strongly ginglymoidal, with a lateral condyle that is slightly wider and more distally expanded lateral than the medial condyle, which are separated by a marked, broad groove (*Figure 10A,B*). A shallow collateral extensor depression is present on the obliquely exposed lateral side of the distal end.

Metacarpal III is slender and slightly shorter than metacarpal II. The bone is straight and closely appressed to metacarpal II over its entire length (*Figure 8*), as in most basal paravian theropods, but unlike the flexed third metacarpal with a well-developed *spatium intermetacarpale* in most modern birds and some basal avialans, such as *Jeholornis* (*Zhou and Zhang, 2002*; *Zhou and Zhang, 2003b*; *Lefèvre et al., 2014*). The proximal end is displaced distally by slightly more than 1 mm from the level of the proximal end of metacarpal I and II, as in oviraptorids (*Longrich et al., 2010*; *Balanoff and Norell, 2012*), dromaeosaurids (*Lü and Brusatte, 2015*), *Archaeopteryx* (*Elzanowski, 2002*; *Mayr et al., 2007*), confuciusornithids (e.g. *Zhang et al., 2008*), enantiornithines (e.g. *Zhang et al., 2013*), and more derived birds. The proximal end has a medially flared, rounded flange that overlaps the lateropalmar edge of metacarpal II and thus reaches a maximal transverse width of 4 mm (*Figure 9*). The flange disappears into the shaft some 4 mm from the proximal end, distal to which the shaft is 2.7 mm wide. The shaft remains of subequal width over most of its length, tapering only very slightly and reaching a minimal width of 2.4 mm some 5 mm from the distal end. Distally, the bone expands slightly again, mainly due to a rounded mediopalmar flange, to a maximal distal width of 2.9 mm. The distal articular surface is not ginglymoidal, but convex transversely, with a palmar depression between the slightly more massive lateral side of the articular surface and the mediopalmar flange (*Figure 10A,B*).

## Digits

The hand of the new specimen shows the typical theropodan phalangeal formula of 2-3-4 (*Figure 8*), with digit II being by far the longest of the three digits, followed by the subequal digit I (65% of the length of digit II) and digit III (64% of the length of digit II). Considering isolated phalanges, phalanx I-1 is the longest manual phalanx, being only very slightly longer (28.5 mm) than phalanx II-2 (28 mm).

Phalanx I-1 is long, slender, and slightly bowed, being convex plantarly (*Figure 10C*). The phalanx seems to be higher than wide (although this might be exaggerated by compression), and its length is approximately 6.5 times its proximal height. The proximal articular end is poorly preserved, but it seems to have been narrow and weakly ginglymoidal. On the palmar side of the proximal end a notable round tubercle probably marks the insertion of a flexor tendon. Directly distal to the proximal articular end, a short, but stout lateropalmar flange seems to have been present (*Figure 10C*). At mid-shaft, the phalanx seems to have a shallow longitudinal lateral groove, similar to, but less developed than in *Ostromia* (*Foth and Rauhut, 2017*); however, as this groove is only apparent in the mid-section of the shaft, and the bone has generally suffered from strong compression, collapsing both the proximal and distal part, some uncertainty remains if this groove represents an original feature or an artefact of preservation. Interestingly, however, such a groove is not present in any of the other manual phalanges, but a similar structure is present in the also apparently uncrushed midshaft of the radius, and we thus tentatively regard this feature as a true character of this phalanx. The distal articular end of phalanx I- one is ginglymoidal, with the palmar side of the articular facet extending considerably further proximally than the plantar side. A well-developed collateral ligament pit is present and displaced plantarly from the mid-height of the distal end.

The ungual of the first digit (*Figure 10D*) is slightly smaller than the ungual of digit II, as in *Anchiornis* (*Hu et al., 2009*) and *Archaeopteryx* (*Wellnhofer, 2008*; *Wellnhofer, 2009*), whereas

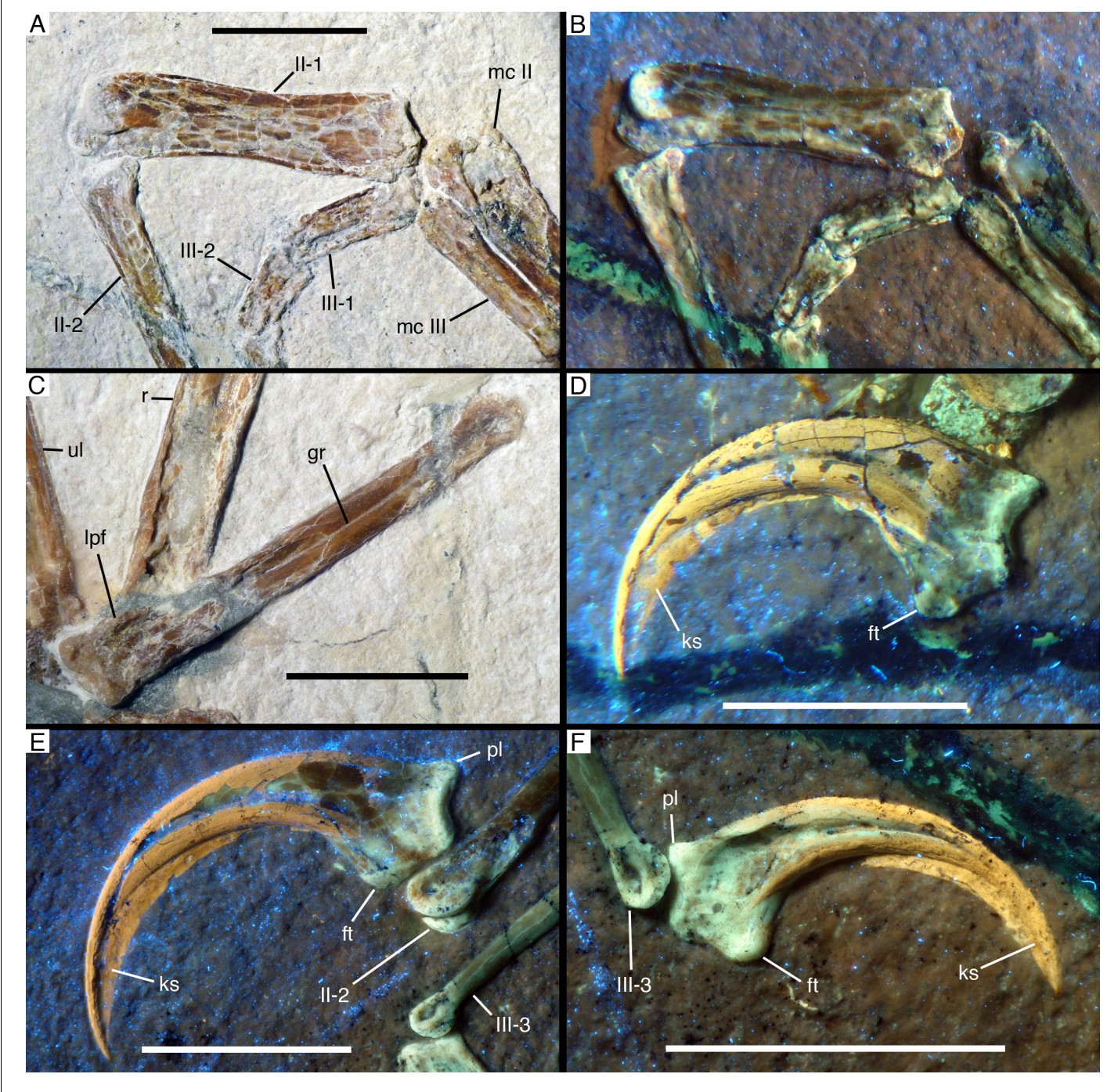

**Figure 10.** Manual phalanges of *Alcmonavis poeschli*. (**A**) and (**B**) Distal ends of metacarpals II and III and proximal phalanges of digits II and III in normal (**A**) and ultraviolet (**B**) light. (**C**) First phalanx of first digit in medial view. (**D**) Ungual phalanx of digit I in medial view under ultraviolet light. (**E**) Ungual phalanx of digit II in medial view under ultraviolet light. (**F**) Ungual phalanx of digit III in lateral view under ultraviolet light. Abbreviations as in *Figure 2*, and: ft, flexor tubercle; gr, groove; ks, keratinous sheath; lpf, lateropalmar flange; pl, proximal lip. Scale bars are 1 cm.

DOI: https://doi.org/10.7554/eLife.43789.012

ungual I is larger than that of the second digit in more basal tetanurans (e.g. *Allosaurus*: *Gilmore, 1920*; oviraptorosaurs: *Clark et al., 1999*; *Balanoff and Norell, 2012*; *Deinonychus*: *Ostrom, 1969*), but also in *Sapeornis* (*Zhou and Zhang, 2003a*) and *Confuciusornis* (*Chiappe et al., 1999*). The ungual of the first digit is strongly curved, with the tip of the bony claw being placed

approximately 5.6 mm below the proximal articular facet, when the latter is oriented perpendicular. The proximal end has a maximal height of 7.2 mm, of which 3.9 mm are accounted for by the proximal articular facet, whereas the rest forms the very strongly developed flexor tubercle. The latter expands downwards from the articular surface to form a rounded palmar tubercle, the distopalmar extremity of which is expanded transversely into a small, triangular lateral tubercle. Distally, the flexor tubercle fades into the palmar margin of the ungual. When the proximal articular end is oriented perpendicular, the plantar margin of the ungual arches slightly upwards in its proximal part, as in most coelurosaurs, but in contrast to more basal theropods. A plantar lip at the proximal end of the ungual is absent. The ungual has a single, more or less centrally placed claw groove, palmar to which the bone is considerably wider transversely than on the plantar side. The palmar margin of the bony ungual is notably flattened, with the claw showing its greatest transverse width where the lateral side flexes into the palmar side, and tapering slightly towards the claw groove. The keratinous sheath of the claw is well preserved and largely hides the distal end of the bony ungual. It extends the length of the claw for approximately one third of the length of the bony element (measuring the maximum straight length of the ungual) and follows the curvature of the latter, so that the distal end of the sheath is placed approximately 9.5 mm below the proximal articular facet when the latter is oriented vertically.

In the second digit, the first phalanx is shorter (c. 81%) than the second phalanx, as in all theropods. However, the first phalanx of the second digit, which is exposed in palmar view, is strikingly robust, being by far the most robust phalanx of the manus (*Figures 8* and *10A,B*), as it is the case in *Sapeornis* (*Provini et al., 2009*; *Gao et al., 2012*), *Confuciusornis* (*Chiappe et al., 1999*) and more derived avialans, whereas this phalanx is only slightly more robust than phalanx II-2 in *Archaeopteryx* (*Mayr et al., 2007*; *Wellnhofer, 2008*; *Wellnhofer, 2009*) and more basal theropods. Thus, with a maximal transverse width of 5.5 mm, the proximal end of phalanx II-1 is more robust than the distal end of its respective metacarpal, and the bone narrows only slightly to a minimal width of 4.4 mm just distal to its mid-length. Although the phalanx is compressed, the preserved width seems to reflect the true width of the element, as indicated by the fitting articulation with metacarpal II. In contrast to the condition in more derived birds, but as in *Sapeornis* (*Provini et al., 2009*) and *Confuciusornis* (*Chiappe et al., 1999*), the phalanx is not flattened but seems to have been rather robust, although it is largely collapsed. The proximal end of the phalanx has two well-developed, concave articular facets separated by a median ridge, fitting with the ginglymoidal distal articular end of the second metacarpal (*Figure 10A,B*). In the shaft, the medial side of the bone is gently concave over its entire length, whereas the lateral side is slightly bulbously convex over its proximal third and becomes only slightly concave distally. The distal articular end of this phalanx is ginglymoidal, being somewhat twisted so that the palmar condyles face slightly lateropalmarly (*Figure 10A,B*), which is furthermore slightly exaggerated by compression. The articular end thus consists of two well-rounded condyles separated by a notable groove. A shallow collateral ligament depression is present on the medial side of the end, being slightly displaced palmarly from the mid-height of the bone.

Phalanx II-2 is long, slender, and slightly flexed, being convex on the plantar side; it is exposed in medial view. The proximal end is 4.4 mm high plantar-palmarly and seems to have been narrow transversely, although this is certainly exaggerated by compression. The proximal articular end is concave. The height of the bone diminishes rapidly directly distal to the articular end, and then more gradually to a minimal value of 2.7 mm approximately two-thirds of the length of the bone from the proximal end. From here, the bone expands slightly again towards the strongly ginglymoidal distal articular end. The latter is very similar to the distal articular end of phalanx I-1 in extending further proximally on the palmar side and having a well-developed, plantarly displaced collateral ligament pit (*Figure 10E*).

The ungual of the second digit (*Figure 10E*) is slightly longer proximodistally than the ungula of digit I, but less strongly curved, so that the distal tip of the bony ungual is placed 5.5 mm below the articular facet. The flexor tubercle is slightly more offset distally from the proximal end than in the first ungual and accounts for c. 3 mm of the maximal proximal height of 6.6. mm. Furthermore, in contrast to ungual I this element shows a well-developed proximal lip on the plantar surface. Otherwise the ungual is similar to that of digit I, in having a transverse expansion of the palmar extremity of the flexor tubercle and a flattened palmar margin. The dorsal margin arches upwards from the

articular end before the claw flexes downward again, as in the first ungual. The ungual sheath is also well preserved and extends to c. 13 mm below the proximal articular surface.

In digit III, the proximal two phalanges are shorter, taken together, than phalanx III-3 (*Figure 8*), as in maniraptoriforms generally (*Rauhut, 2003*). Of the two proximal phalanges, phalanx II-1 is considerably longer (c. 150%) than phalanx II-2, but both are poorly preserved (*Figure 10A,B*). Both phalanges are strongly compressed and collapsed, but the transverse width of phalanx III-1 can be estimated to be approximately 1.5 mm, or only about 25–30% that of the maximal width of phalanx II-1. Both phalanges seem to have been ginglymoidal, but nothing can be said about any morphological details. Phalanx III-3 is long and slender but straight, not flexed as it is the case in phalanges I-1 and II-2. The proximal end is poorly preserved but was approximately 2.5 mm high plantar-palmarly. From this end, the phalanx gradually tapers to a minimal height of 0.9 mm just proximal to the expansion of the ginglymoidal distal articular end. The latter is again generally similar to the distal ends of phalanges I-1 and II-2, being more narrow on the plantar than on the palmar side (*Figure 10E*).

The ungual of digit III (*Figure 10F*) is by far the smallest of the manual unguals, as in most theropods. It is also less markedly curved than the other unguals, with the tip of the bone being placed approximately 2.4 mm below the articular facet. The flexor tubercle is proximally placed and similarly pronounced as in the first ungual and accounts for c. 2.2 mm of the maximal proximal height of 4.6 mm of the bone. In other characters, such as the upward arching proximal flexure of the claw, the singly claw groove, and the flattened palmar margin, the ungual is similar to the other manual unguals. This is also true for a transverse expansion of the distopalmar extremity of the flexor tubercle, further indicating that this represents the original condition and does not stem from compression or deformation. As in the other unguals, the sheath extends the ungual for slightly more than one third of the length of the bony element, and its tip is placed c. 5.2 mm below the proximal articular facet.

## Discussion

### Taxonomic identification of SNSB-BSPG 2017 I 133

Traditionally, all paravian specimens from the late Kimmeridgian - early Tithonian laminated limestones of southern Germany have been identified as *Archaeopteryx*, and this would thus be an obvious identification for the new specimen as well. However, as noted in the introduction, recent discoveries of basal paravian and even avialan theropods, also from the Jurassic, have made the distinction of *Archaeopteryx* from other basal avialans (and some small, more basal paravians, such as *Microraptor*) difficult, and there is no reason for an *a priori* assumption that all paravian specimens from this area should represent a single genus or even a single lineage (*Foth and Rauhut, 2017*). Unfortunately, neither the recent diagnosis of the genus *Archaeopteryx* by *Rauhut et al. (2018)* nor that by *Kundrát et al. (2019)* includes any forelimb characters, and due to the great similarity of the forelimbs of many non-ornithothoracan paravians, detailed comparisons are necessary to approach the taxonomic identity of SNSB-BSPG 2017 I 133. These comparisons are further complicated by the fact that the forelimb bones in almost all specimens of *Archaeopteryx*, and many other relevant taxa known from flattened specimens in matrix slabs, are exposed in dorsal view, while they are exposed in ventral view in the current specimen.

Another problem in comparing the new specimen with specimens of *Archaeopteryx* is that it is considerably larger in size than any of the other specimens (*Table 1*, see *Mayr et al., 2007* and *Rauhut et al., 2018* for comparison). Thus, based on comparisons of the length of the ulna, the only long bone that can be measured with certainty, the new urvogel specimen is more than 220% of the size of the smallest known *Archaeopteryx*, the Eichstätt specimen, and still c. 111% the size of the largest specimen, the Solnhofen specimen. Compared with the only *Archaeopteryx* known from the Mörnsheim Formation, the ulna of SNSB-BSPG 2017 I 133 is almost 175% of this specimen. Thus, possible allometric and/or ontogenetic changes have also to be taken into account. This is not only true for proportions, as muscle insertion areas often also become more conspicuous with age and size in vertebrates (e.g. *Hübner, 2010*).

As far as proportions can be evaluated, the new specimen is generally closely comparable to specimens that can certainly be identified as *Archaeopteryx*, especially in the ratio of the (estimated)

**Table 1.** Measurements of the right forelimb of *Alcmonavis poeschli*.
All measurements in millimetres. Length of unguals is given as maximal length measured in a straight line from proximal articulation to tip.

| Element | Length | Element | Length |
|---|---|---|---|
| Humerus | 90 (estimated) | Phalanx II-1 | 22.8 |
| Ulna | 82 | Phalanx II-2 | 28 |
| Metacarpal I | 7.1 (estimated) | Ungual II | 17.5 |
| Metacarpal II | 40.9 | Phalanx III-1 | 8.8 |
| Metacarpal III | 36.8 | Phalanx III-2 | 5.9 |
| Phalanx I-1 | 28.5 | Phalanx III-3 | 17.9 |
| Ungual I | 16 | Ungual III | 11 |

DOI: https://doi.org/10.7554/eLife.43789.013

length of humerus versus ulna, ulna versus maximal length of metacarpus, and metacarpal II versus length of various phalanges. However, the significance of this similarity is unclear, as these proportions are also comparable in a wide variety of other basal paravians, including *Anchiornis*, *Sapeornis*, and, at least for several of these ratios, also *Microraptor* (**Hwang et al., 2002**; **Pei et al., 2014**). Interestingly, however, differences in proportions are found in a few ratios (see *Table 2*), most notably in the length of the manual unguals. When compared to the ulna length, manual unguals are relatively smaller than in specimens of *Archaeopteryx*. This is most marked in comparison with unguals I and III of the largest specimen of *Archaeopteryx*, the Solnhofen specimen: although, as noted above, the ulna of this specimen is about 10% shorter than that of SNSB-BSPG 2017 I 133, its unguals I and III are even slightly longer than in the new specimen.

Based on a one-sample t test (see *Table 2*), the new specimen differs significantly from *Archaeopteryx* in the following ratios: manual phalanx I-1 vs. ulna, manual phalanx II-1 vs. manual phalanx I-1, manual phalanx II-2 vs. manual phalanx I-1, manual phalanx III-1 vs. manual phalanx I-1, manual ungual II vs. manual phalanx II-1, and manual digit I vs. manual digit II. When the juvenile Eichstätt specimen is excluded the ratios of manual phalanx III-2 vs. ulna, and manual phalanx III-2 vs. manual

**Table 2.** Comparison of skeletal proportions of specimens of *Archaeopteryx* and the new Mühlheim specimen (bold).

| Specimen | PI-1/Ulna | Piii-1/PI-1 | Pii-1/PI-1 | Pii-2/PI-1 | Digit I/Digit II | UI/Ulna | Uii/PII-1 | PIII-2/Ulna | Piii-2/PIII-1 |
|---|---|---|---|---|---|---|---|---|---|
| Eichstätt* | 0.422 | 0.246 | 0.656 | 0.942 | NA | NA | NA | 0.060 | 0.458 |
| Thermopolis | 0.383 | 0.250 | 0.656 | 0.954 | 0.699 | 0.210 | 0.922 | 0.083 | 0.875 |
| Munich | 0.377 | 0.298 | 0.625 | 0.900 | 0.751 | 0.204 | 0.840 | 0.075 | 0.800 |
| Berlin | 0.383 | NA | 0.712 | 0.902 | 0.682 | 0.208 | 0.915 | 0.071 | 0.625 |
| London | NA | 0.282 | NA | NA | NA | NA | NA | NA | NA |
| Solnhofen | 0.378 | 0.312 | 0.679 | 0.964 | 0.713 | 0.226 | 0.879 | 0.082 | 0.772 |
| Daiting | 0.382 | NA | NA | NA | NA | NA | NA | NA | NA |
| Altmühl | 0.373 | 0.275 | 0.798 | 0.841 | 0.705 | 0.227 | 0.806 | 0.085 | 0.828 |
| Schamhaupten | NA | NA | NA | NA | NA | NA | NA | NA | NA |
| Maxberg | 0.403 | 0.280 | 0.760 | 0.880 | NA | NA | NA | 0.081 | 0.714 |
| Ottmann and Steil | 0.363 | 0.280 | 0.733 | 0.933 | NA | NA | NA | 0.081 | 0.794 |
| **Mühlheim** | **0.348** | **0.309** | **0.800** | **0.982** | **0.652** | **0.195** | **0.768** | **0.072** | **0.670** |
| *T* value* | 6.323 | −3.996 | −4.684 | −4.598 | NA | NA | NA | 1.783 | 1.330 |
| *p* value* | **0.000** | **0.005** | **0.002** | **0.002** | NA | NA | NA | 0.118 | 0.225 |
| *T* value | 8.038 | −3.642 | −3.988 | −4.373 | 5.068 | 4.170 | 4.722 | 4.146 | 3.324 |
| *p* value | **0.000** | **0.011** | **0.007** | **0.005** | **0.007** | **0.014** | **0.009** | **0.006** | **0.016** |

DOI: https://doi.org/10.7554/eLife.43789.014

phalanx III-1 are also significant different from each other. A further significant difference from specimens of *Archaeopteryx* might be the length of metacarpal I, which seems to be considerably shorter in the new specimen, under the assumption that the position of the proximal end of the first manual phalanx indicates the length of this bone. However, as the distal end of metacarpal I is not preserved, this cannot be established with any certainty.

Likewise difficult to establish are probable differences in relative robusticity of structures in comparison with *Archaeopteryx*, mainly because of the strong compression of the new specimen. One striking feature of the new specimen is the width of the deltopectoral crest of the humerus, which seems to considerably exceed the width of the humeral shaft, similar to *Confuciusornis* (*Chiappe et al., 1999*) and *Ichthyornis* (*Clarke, 2004*). This is an unusual feature not seen in specimens of *Archaeopteryx*, but some uncertainty remains due to the strong compression of the bone. Also unusual in the humerus is the angle at which the proximal part that bears the deltopectoral crest is offset from the distal shaft (*Figure 11*). This angle is below 30° in dromaeosaurids, such as *Microraptor* (*Hwang et al., 2002*; *Pei et al., 2014*), *Zhenyuanlong* (*Lü and Brusatte, 2015*), and *Deinonychus* (*Ostrom, 1969*), and most specimens of *Anchiornis* (*Hu et al., 2009*; *Pei et al., 2017*), and varies between 30° and 33° in specimens of *Archaeopteryx* (*Figure 11B–D*; see *Wellnhofer, 2008*, *Wellnhofer, 2009*). However, this angle is 38° in the Mühlheim specimen, which is close to *Confuciusornis* (36°: SNSB-BSPG 1999 I 15; 38°: JME 1996/15, 1997/1; *Figure 11H*) and some other more derived avialans such as *Sulcavis* (*O'Connor et al., 2013*), *Archaeorhynchus* (e.g. *Zhou et al., 2013*), *Yanornis* (*Zhou and Zhang, 2001*) or *Gansus* (*Wang et al., 2016*).

In the manus, the extreme robusticity of metacarpal II in comparison with the other metacarpals is striking. As noted in the description, the proximal part of this metacarpal is almost twice as wide as the proximal articular surface of metacarpal I. However, this metacarpal is not only robust in comparison to the other metacarpals, but also in itself: whereas the length of metacarpal II exceeds ten times the maximal width of the bone in specimens of *Archaeopteryx*, it is only around 8.2 times the maximal width of this element in SNSB-BSPG 2017 I 133 (see *Figure 12C,E*). As the metacarpus is preserved in articulation, it seems very unlikely that this robusticity of metacarpal II is entirely due to preservation, and thus probably represents a true difference of the new specimen from specimens of *Archaeopteryx* (*Figure 12*). On the other hand, this feature resembles the condition of many basal birds from the Jehol Group, for example *Jeholornis* (*Zhou and Zhang, 2002*; *Lefèvre et al., 2014*) and *Archaeorhynchus* (*Zhou et al., 2013*), which highlight the transition of an individual metacarpal II to the major element of the fused capometacarpus of modern birds.

Another element in the manus that is remarkably robust is the first phalanx of the second digit. In non-avialan paravians and specimens that can securely be referred to *Archaeopteryx* (e.g. *Figure 12A–C*), this phalanx is only slightly more robust than phalanx II-2 or III-1 (e.g. 1.25 times the width of phalanx II-2 at the proximal shaft in the Berlin specimen), but it is more than 1.7 times the width of the widest part of phalanx II-2 and more than twice the width of phalanx II-1 in the Mühlheim specimen. This pronounced robusticity of phalanx II-1 in comparison with *Archaeopteryx* (and more basal paravians) probably represents an apomorphic character shared by SNSB-BSPG 2017 I 133 and more derived avialans, as a widened phalanx proximalis digiti majoris is a general character of avialans more derived than *Archaeopteryx*, and is present e.g. in basal forms, such as *Jeholornis* (*Lefèvre et al., 2014*), *Sapeornis* (*Yuan, 2008*; *Provini et al., 2009*), *Chongmingia* (*Wang et al., 2016*), and *Confuciusornis* (*Figure 12D*; *Chiappe et al., 1999*). The phalanx in the Mühlheim specimen is plesiomorhic in comparison to most more derived avialans (i.e., Ornithothoraces) in that it is not dorsoventrally flattened, but seems to be rather robust also in this plane. In this respect it is similar to the robust phalanx in *Jeholornis* (*Lefèvre et al., 2014*), *Sapeornis* (*Yuan, 2008*; *Provini et al., 2009*) and *Confuciusornis*. In the latter, this element is dorsoventrally robust at least along its anterior edge, whereas the posterior edge is somewhat flattened (*Figure 12D*; SNSB-BSPG 1999 I 15, JME 1997/1, 2005/1).

Apart from these morphometric differences, there are also several qualitative characters that differ between the new specimen and specimens of *Archaeopteryx*. One of these characters concerns the insertion of the *m. pectoralis* on the deltopectoral crest of the humerus. This facet is not especially marked, or only indicated by slight thickening of the apex of the crest in most non-avialan theropods. In *Archaeopteryx*, the anterior side of the deltopectoral crest is only exposed in the London, Thermopolis and, partially, the Maxberg specimens (pers. obs.; see *de Beer, 1954*; *Heller, 1959*; *Mayr et al., 2007*; *Wellnhofer, 2008*; *Wellnhofer, 2009*). Although the edge of the

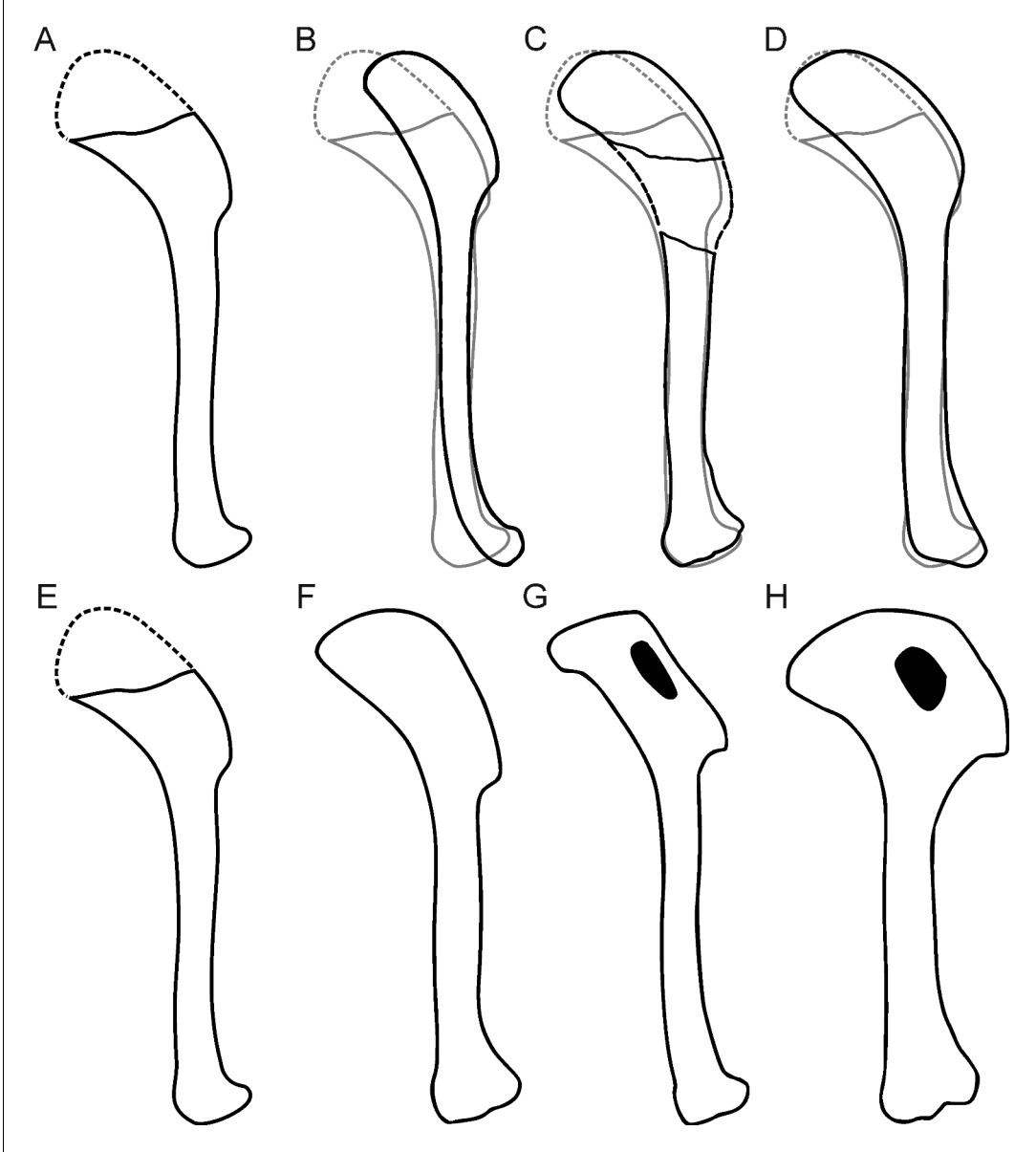

**Figure 11.** Comparison of humeral shape in some Mesozoic birds. (**A, E**) *Alcmonavis*. (**B–D**) *Archaeopteryx* with humeral shape of *Alcmonavis* shown in grey. (**B**) Berlin specimen. (**C**) Solnhofen specimen. (**D**) Daiting specimen. (**F**) *Jeholornis*. (**G**) *Sapeornis*. (**H**) *Confuciusornis*. (**B, C**) modified after *Wellnhofer, 2008*. (**F–H**) modified after *Wang et al., 2016*.
DOI: https://doi.org/10.7554/eLife.43789.015

deltopectoral crest is slightly damaged in the London and Thermopolis specimens, it can be established that these specimens follow the general theropodan condition of not showing a marked facet (*Figure 13*). In contrast, SNSB-BSPG 2017 I 133 shows a well-developed, anteromedially inclined, elongate oval facet for the insertion of the *m. pectoralis* on the deltopectoral crest (*Figure 3*). Again, this character is present in many basal avialans from the Jehol group (e.g. *Figure 14*) and seems to be a derived character in comparison with *Archaeopteryx* (see below).

Another striking feature of SNSB-BSPG 2017 I 133 is the development of the *tuberculum bicipitale radii* as a raised crest on the proximal radius. However, due to preservation the exact development of this tuberculum in specimens of *Archaeopteryx* is difficult to establish. Several specimens, including the Munich, Altmühltal and Ottmann and Steil ("chicken wing") specimens show that this

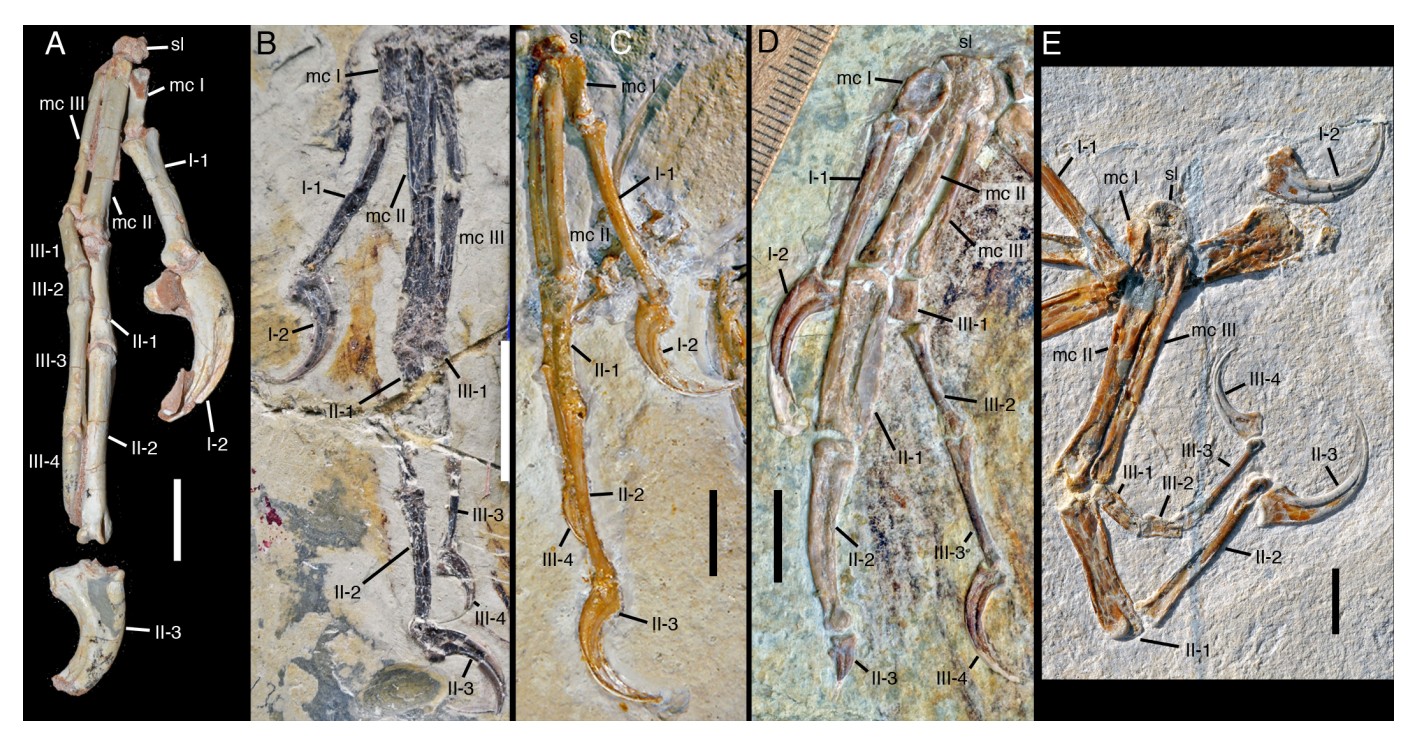

**Figure 12.** Mani of several paravian theropods for comparison with *Alcmonavis*. (**A**) Right manus of *Velociraptor mongoliensis* (IGM 100/982) in plantar view. (**B**) Left manus of *Microraptor gui* (IVPP V 13352) in plantar view. (**C**) Right manus of the Thermopolis specimen of *Archaeopteryx* in planto-medial view. (**D**) Left manus of *Confuciusornis sanctus* (JME 2005/1) in palmar view. (**E**) Right manus of *Alcmonavis poeschli* in palmar view. Abbreviations as in *Figure 2*. Scale bars are 20 mm (**A**) and 10 mm (**B–E**).

DOI: https://doi.org/10.7554/eLife.43789.016

tuberculum is present in this taxon. Although a direct comparison of this structure is difficult due to preservation, these specimens seem to show a rounded to triangular expansion that is different from the crest-like, more rectangular tubercle in the new specimen. In those avialan specimens from the Jehol Group where this characters can be evaluated it varies between a triangular and crest-like state (*Chiappe et al., 1999*; *Provini et al., 2009*; *Zhou et al., 2013*).

If the interpretation of longitudinal furrows on the radius and phalanx I-1 as original features is correct, these are further differences from *Archaeopteryx*. As noted above, such furrows are observed in some other paravian theropods in variable elements (see e.g. *Chiappe and Walker, 2002*; *Sanz et al., 2002*; *Hu et al., 2015*; *Foth and Rauhut, 2017*; *Xu et al., 2017*), but the combination of such furrows in the radius and only one manual phalanx might be unique for SNSB-BSPG 2017 I 133. However, as such features might be easily overlooked, more studies of these morphologies are needed.

Finally, apart from being relatively smaller, the manual unguals also show differences from those of specimens of *Archaeopteryx*. Especially the shape, position and prominence of the flexor tubercle seems to differ. In general the flexor tubercles in the new specimen seem to be placed slightly more proximally than in specimens of *Archaeopteryx* and the tubercles are more pointed, that is the angle between their proximal and distal margin are sharper; this is especially marked in manual ungual II. Likewise, the transverse distopalmar expansion of the flexor tubercle, which is present in all three unguals of the Mühlheim specimen, is not present in specimens of *Archaeopteryx*.

The comparison with the anchiornithid *Ostromia* (*Foth and Rauhut, 2017*) is more difficult due to the fragmentary nature of both specimens. Most of the forelimb bones of *Ostromia* are actually preserved as imprints, so that only parts of the manus can be used for comparison. Like *Ostromia*, SNSB-BSPG 2017 I 133 seems to possess a longitudinal furrow along the manual phalanx I-1, a character that is also shared with *Sinornithosaurus* and *Jianianhualong* (*Foth and Rauhut, 2017*;

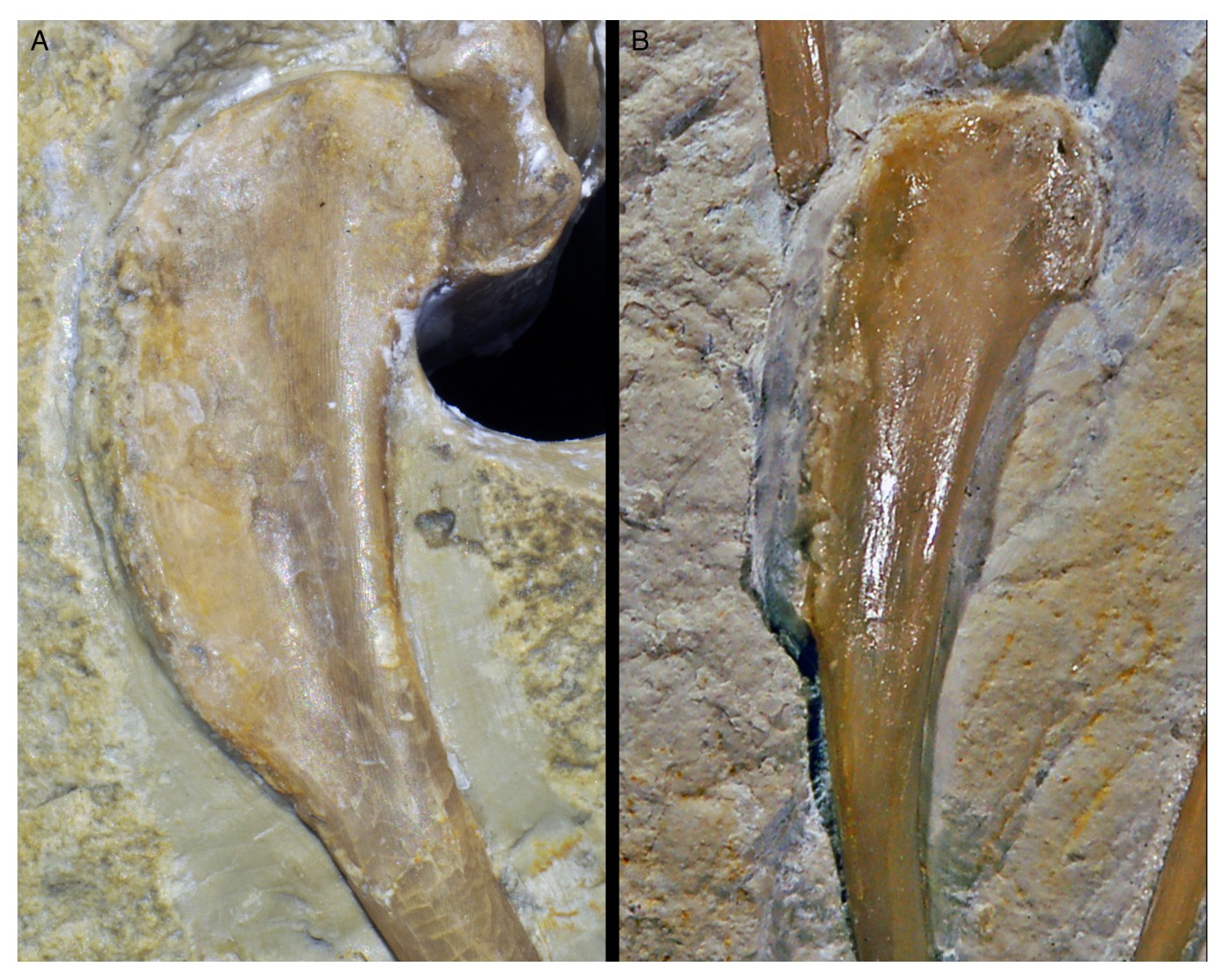

**Figure 13.** Proximal part of the humerus of specimens of *Archaeopteryx* in anteromedial view, showing the lack of a pronounced facet for the pectoralis muscle in this taxon. (**A**) London specimen. (**B**) Thermopolis specimen.
DOI: https://doi.org/10.7554/eLife.43789.017

*Xu et al., 2017*). However, *Ostromia* shows similar grooves also in manual phalanx III-3, while the corresponding element is smooth in *Alcmonavis*. Striking differences between *Alcmonavis* and *Ostromia* are present in the size and shape of the manual unguals. When compared to the length of the manual phalanx I-1, the first ungual of *Ostromia* is much smaller than that of *Alcmonavis*. In contrast, the third ungual is much smaller in *Alcmonavis* when compared to the size of the first ungual, while in *Ostromia* they almost have the same size. While some basal avialans like *Jeholornis* and *Sapeornis* from the Jehol Group show enlarged manual unguals, they also show a clear size reduction from manual ungual I to III (*Zhou and Zhang, 2002*; *Zhou and Zhang, 2003a*; *Provini et al., 2009*; *Gao et al., 2012*). The flexor tubercles of the manual unguals in *Ostromia* are much more distally displaced than in SNSB-BSPG 2017 I 133 (but also *Archaeopteryx*). As described above, the flexor tubercles of SNSB-BSPG 2017 I 133 are very prominent and pointed ventrally, while in *Ostromia* they are relatively low, forming a plateau-like ventral apex (see *Wellnhofer, 2008*: Figure 5.79B, C; *Foth and Rauhut, 2017*: Figures 3B and 4). Furthermore, like *Archaeopteryx*, *Ostromia* lacks the transverse distopalmar expansion of the flexor tubercle and a notably flattened palmar margin of

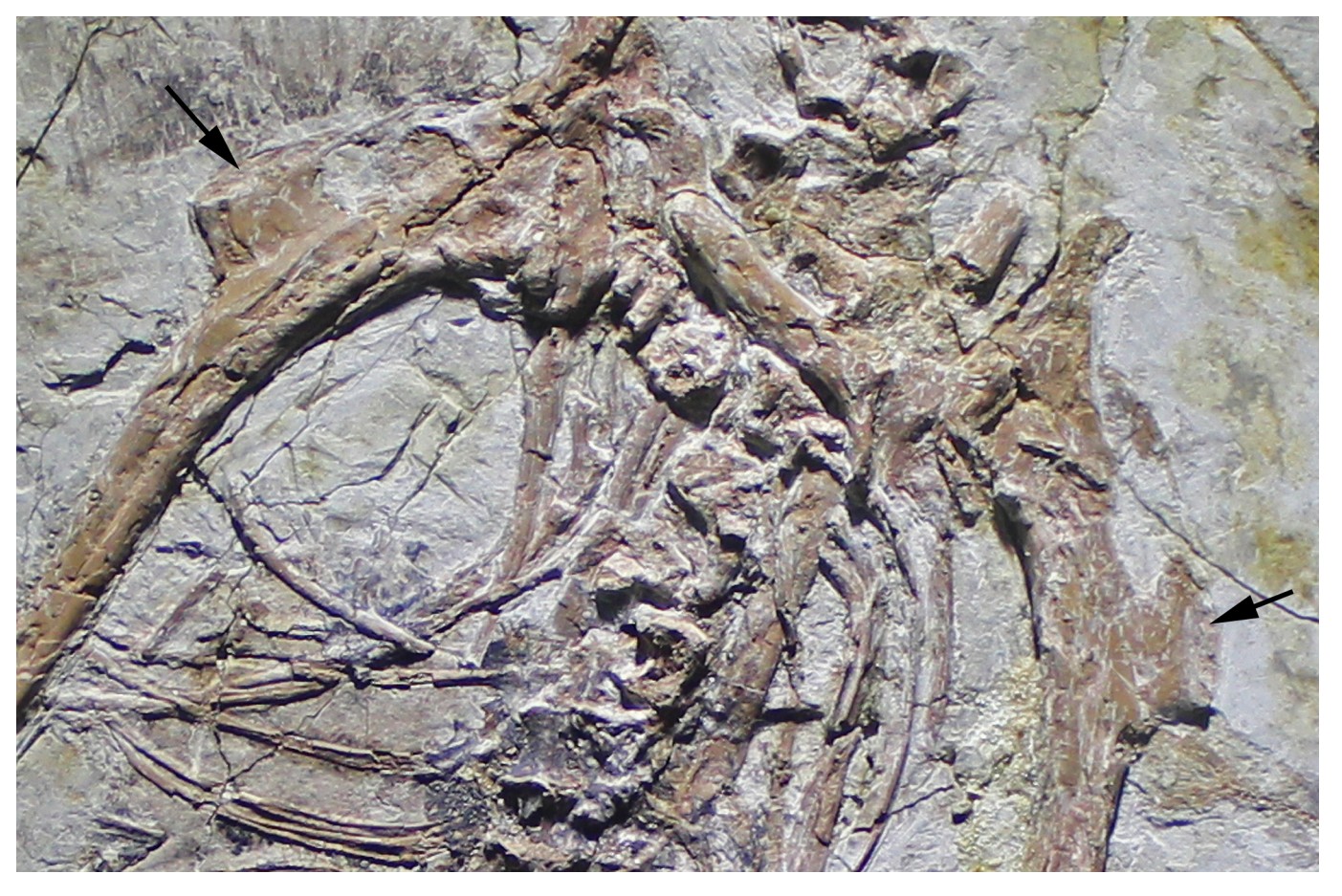

**Figure 14.** Shoulder region and proximal ends of both humeri of *Sapeornis* (JZT-DB 0047), showing the enlarged and medially inclined facet for the insertion of *m. pectoralis* (arrows).
DOI: https://doi.org/10.7554/eLife.43789.018

the bony ungual. While the manual unguals of *Confuciusornis*, *Jeholornis* and *Sapeornis* still bear prominent flexor tubercles (**Chiappe et al., 1999**; **Zhou and Zhang, 2002**; **Zhou and Zhang, 2003b**), they become reduced within Ornithothoraces (e.g., **Zhou et al., 2013**).

In summary, despite the overall similarity and very similar proportions, the new specimen shows numerous small differences from *Archaeopteryx*, precluding a referral to this taxon. Several characters, including the markedly concave proximal articular surface of the ulna, the very massive phalanx II-1 and the marked, anteromedially inclined facet for the attachment of the m. pectoralis in the humerus, are shared with more derived avialans, and indicate that the Mühlheim specimen represents a third, phylogenetically slightly more crownward taxon of avialans from the Tithonian limestones of southern Germany. From the description and comparisons of the specimen it is furthermore clear that SNSB-BSPG 2017 I 133 also cannot be referred to *Ostromia* or to any other known theropod taxon. We thus opt to describe this rather incomplete specimen as a new genus and species of 'Urvogel'.

## Systematic palaeontology

Theropoda *Marsh, 1881*
Maniraptora *Gauthier, 1986*
Avialae *Gauthier, 1986*
*Alcmonavis poeschli* gen. et sp. nov.
urn:lsid:zoobank.org:act:668F42B6-5BDC-4ADF-B271-36C6A43C7DB3

## Etymology

From *Alcmona*, the old Celtic name of the Altmühl River, which flows through the principal region in which the famous 'Solnhofen limestones' are exposed, and avis, from the Greek 'aves' for bird. The species name honours Roland Pöschl, who leads the excavations at the Schaudiberg and found the specimen.

## Holotype

SNSB-BSPG 2017 I 133, an almost complete, partly disarticulated skeleton of the right wing (*Figure 2*, see Tab. 1 for measurements of SNSB-BSPG 2017 I 133).

## Locality and horizon

Old Schöpfel Quarry at the Schaudiberg, Mühlheim, close to Mörnsheim, Bavaria. Mörnsheim Formation, *moernsheimensis* ammonite horizon of the *Hybonotum* zone of the Early Tithonian. The specimen comes from a thin layer of marly laminated limestone some 6 m above the contact with the underlying Altmühltal Formation.

## Differential diagnosis

*Alcmonavis poeschli* differs from all other theropods (including birds) in the following combination of characters: humerus with large deltopectoral crest, with a maximal expansion that exceeds the width of the humeral shaft; proximal part of humerus strongly angled at approximately 38° in respect to distal shaft; ulna with well-defined, single, oval, concave proximal cotyla and small lateral tubercle; distal end of ulna slightly asymmetrically expanded; large, crest-like biceps tubercle on the proximal radius; longitudinal groove along the medial side of the radial shaft; metacarpal II considerably more robust than metacarpal I and III; phalanx I-1 with longitudinal groove; phalanx II-1 very robust, but with rounded, rather than flattened cross-section; phalanx II-1 slightly twisted; manual unguals with strongly developed and palmarly transversely expanded flexor tubercles.

## Phylogenetic position of *Alcmonavis*

The phylogenetic analysis (see Materials and methods) resulted in more than 99,999 trees with a length of 2690 steps. The strict consensus (*Figure 15—figure supplement 1*) is rather well resolved and includes monophyletic Maniraptora, Paraves and Avialae with equivalent taxonomic contents to other recent analyses. Areas with lack of resolution include a polytomy between therizinosauroids, oviraptorosaurs and paravians, two larger polytomies at the base of Deinonychosauria, and a polytomy at the base of Ornithothoraces, as well as minor polytomies in the higher nodes or Alvarezsauridae, Oviraptorosauria, and Dromaeosauridae. Reduced consensus methods recovered a number of problematic taxa (*Albinykus*, *Byronosaurus*, *Balaur*, *Citipati*, *Hesperonychus*, *Jinfengopteryx*, *Pyroraptor*, *Xixiasaurus*, *Yixianosaurus*, and a *Vorona-Liaoningornis* clade), the a postiori pruning of which further increased resolution. In contrast to other recent iterations of this matrix (*Foth et al., 2014*; *Foth and Rauhut, 2017*), oviraptorosaurs and therizinosaurs were found in a monophyletic clade in the reduced consensus tree (*Figure 15*, *Figure 15—figure supplement 1*), as in several earlier phylogenetic analyses (e.g. *Makovicky and Sues, 1998*; *Holtz, 2000*; *Rauhut, 2003*; *Turner et al., 2011*), and the analysis recovered a monophyletic Deinonychosauria, including Troodontidae and Dromaeosauridae as sister groups, as in most analyses of coelurosaur interrelationships. *Epidexipteryx*, often considered to be a basal paravian (e.g. *Turner et al., 2012*; *Godefroit et al., 2013a*; *Godefroit et al., 2013b*; *Xu et al., 2015*) or even avialan theropod (e.g. *Xu et al., 2011*; *Foth et al., 2014*) is here recovered as a basal oviraptorosaur, as in *Agnolín and Novas, 2013*. As originally proposed by *Csiki et al., 2010*, *Turner et al., 2012* and *Brusatte et al., 2013*, *Balaur* is placed within Dromaeosauridae in the current analyses and not at the base of Avialae (see *Godefroit et al., 2013a*; *Foth et al., 2014*; *Cau, 2018*; *Foth and Rauhut, 2017*). The controversial Late Jurassic paravians *Anchiornis*, *Xiaotingia*, *Eosinopteryx*, *Pedopenna*, and *Ostromia* were found as basal avialans (as in *Godefroit et al., 2013b*; *Foth et al., 2014*; *Foth and Rauhut, 2017*) but a monophyletic Anchiornithidae (as defined by *Xu et al., 2016*; see also *Foth and Rauhut, 2017*) is restricted to the genera *Eosinopteryx*, *Ostromia* and *Anchiornis*, whereas *Pedopenna* and *Xiaotingia* form sister taxa just basal to *Archaeopteryx*. The new taxon, *Alcmonavis*, was found crownwards to *Archaeopteryx*, thus representing the most derived avialan known from the Jurassic so far. The

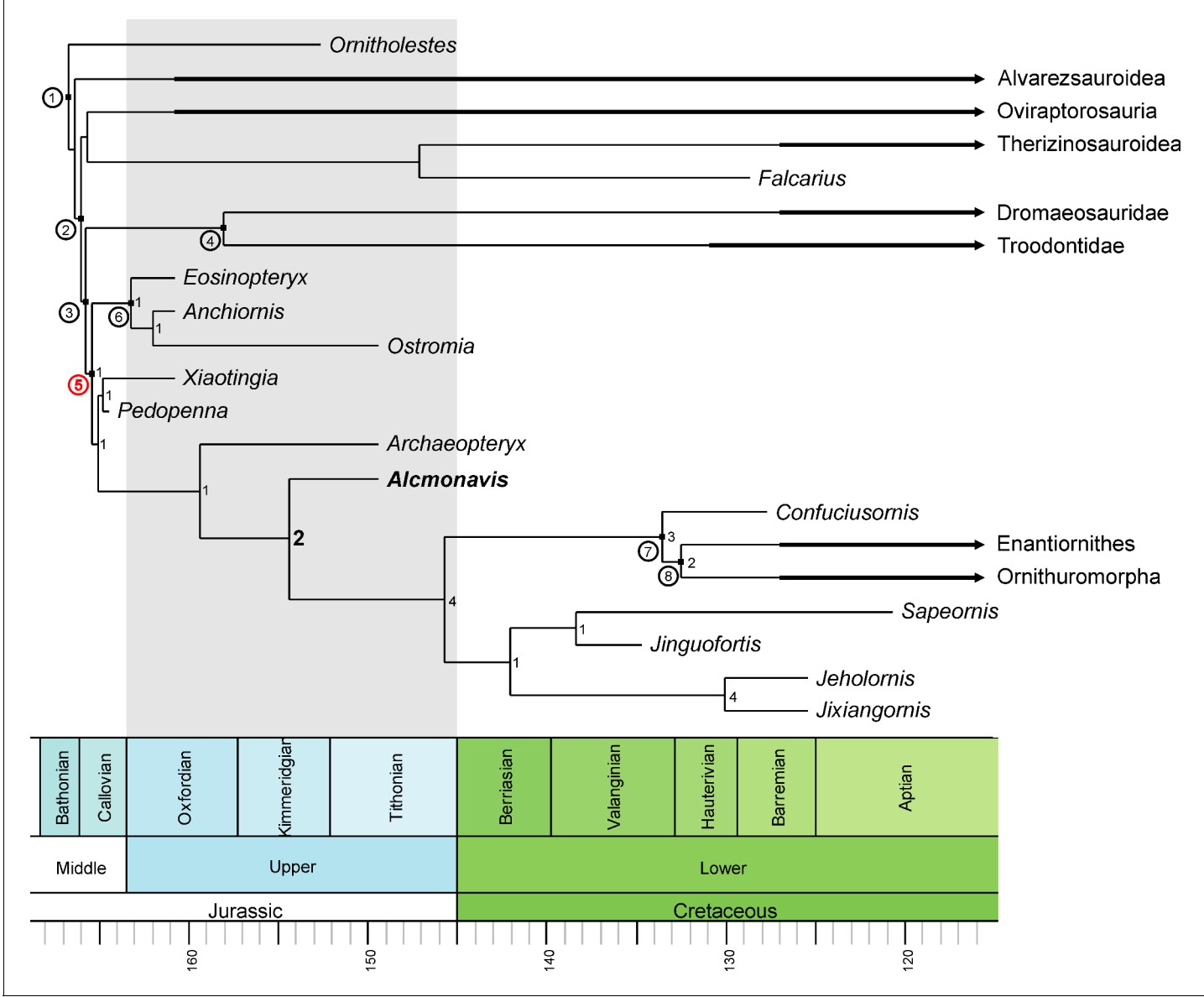

**Figure 15.** Phylogenetic position of *Alcmonavis poeschli*. Time-calibrated, simplified reduced consensus tree resulting from an analysis of 136 taxa scored for 565 characters under equally-weighted parsimony (see text and supplementary material for details). Nodes with circuit numbers: 1, Maniraptora; 2, Pennaraptora; 3, Paraves; 4, Deinonychosauria; 5, Avialae; 6, Anchiornithidae; 7, Pygostylia; 8, Ornithothoraces. Bremer supports are shown for the clade Avialae.

DOI: https://doi.org/10.7554/eLife.43789.019

The following figure supplements are available for figure 15:

**Figure supplement 1.** Full strict consensus tree of the unweighted analysis with Bremer support and Bootstrap values for clades with a support of 50% or more indicated.
DOI: https://doi.org/10.7554/eLife.43789.020

**Figure supplement 2.** Full reduced consensus tree of the unweighted analysis.
DOI: https://doi.org/10.7554/eLife.43789.021

**Figure supplement 3.** Strict consensus tree of the analysis using implied weights with Bremer support and Bootstrap values for clades with a support of 50% or more indicated.
DOI: https://doi.org/10.7554/eLife.43789.022

phylogenetic position of *Alcmonavis* is supported by four synapomorphic characters: char. 217, the distal humeral condyles are positioned on anterior surface (1); char. 562, the attachment for *m. pectoralis* on the deltopectoral crest of the humerus is marked as an elongate oval, anteromedially inclined facet on the mediodistal surface of the deltopectoral crest (1); char. 563, the proximal articular surface of the ulna is developed as an oval concavity with slightly raised rims (1); and char. 565, manual phalanx II-1 is strongly broadened, more than 1.5 times the width of phalanx II-2 and phalanges of digit III (1).

Clade support is low for most clades, as is expected in a matrix with numerous very incomplete taxa and an average amount of missing codings of c. 60%. However, whereas most clades have Bremer support values of 1, the phylogenetic position of *Alcmonavis* is supported by a Bremer value of 2 (*Figure 15*, 15—figure supplement 1). In order to evaluate support for some of the results relevant to the question of bird origins, we ran several constrained analyses. Both the monophyly of Anchiornithidae as proposed by *Foth and Rauhut, 2017* and a position of troodontids closer to Avialae than to dromaeosaurids only requires one additional step, so neither of these possibilities can currently be excluded. Placing anchiornithids in Troodontidae, as argued by *Hu et al., 2009* and subsequent authors, requires at least 10 additional steps, and placing both anchiornithids and *Archaeopteryx* in Deinonychosauria, as proposed by *Xu et al., 2011* leads to trees that are 21 steps longer than the most parsimonious trees, making this arrangement rather unlikely. Concerning the phylogenetic position of *Alcmonavis*, both a placement below *Archaeopteryx*, and as sister taxon to the latter require four additional steps. Given that only 79 of 565 characters (less than 14%) can be coded for the new taxon, this difference indicates that the position retrieved for *Alcmonavis* is rather robust.

The implied weight analysis retrieved 405 equally parsimonious trees with a score of 114.762. The general results of this weighted analysis are similar to those obtained from the analysis under equal weights (*Figure 15—figure supplement 3*), with one notable exception: therizinosaurs are here recovered as the most basal clade of maniraptorans, followed by alvarezsaurids and then oviraptorosaurs, as in *Senter, 2007*. Most importantly, however, the weighted analysis supports the relationships between anchiornithids, *Archaeopteryx*, and *Alcmonavis*.

## Implications for the early evolution of the avian wing

*Alcmonavis* shows several notable characters that might help to elucidate the early steps of the osteological evolution of the bird wing. Current discussions of the evolution of flight capabilities have focused on feather evolution and arrangement (e.g. *Clarke et al., 2006*; *Chiappe et al., 2014*; *Foth et al., 2014*; *O'Connor et al., 2013*; *O'Connor et al., 2016*; *O'Connor and Chang, 2015*; *Sullivan et al., 2017*; *Saitta et al., 2018*), whereas the flight musculature and its osteological correlates has received comparatively little attention recently, with the exception of the study of the shoulder girdle and supracoracoideus muscle of Mesozoic birds by *Mayr, 2017*.

The identification of several muscle attachment areas in the forelimb bones of *Alcmonavis* has implications for our understanding of the early evolution of avialan flight musculature. In recent birds, the most important flight muscles are the *m. pectoralis* (also named *m. pectoralis major* or *m. pectoralis superficialis*), which is the main muscle in the downstroke of the wing, and the *m. supracoracoideus* (*m. pectoralis minor; m. pectoralis profundus*), which lifts the forelimb and has an important role in the rotation of the humerus (*Dial, 1992*; *Ostrom et al., 1999*; *Baier et al., 2007*; *Biewener, 2011*; *Tobalske, 2016*). The supracoracoideus muscle attaches on the anterodorsal edge of the proximal humerus (on the tuberculum dorsale; *Baumel and Witmer, 1993*; see also *Jasinoski et al., 2006*); as this part of the humerus is not preserved in *Alcmonavis*, nothing can be said about this muscle in this taxon. More general discussions of the evolution of the supracoracoideus muscles can be found in *Ostrom, 1976b*, *Ostrom et al., 1999*, *Baier et al., 2007* and *Mayr, 2017*. The deltopectoral crest and attachment area for the pectoralis muscle, is, however, generally well-preserved in the new taxon.

Dinosaurs in general are noteworthy for having a well-developed deltopectoral crest on anterolateral side of the proximal end of the humerus. This crest serves as attachment site for several proximal forelimb muscles that extend from the shoulder girdle to the humerus. In crocodiles, these muscles are mainly the *m. coracobrachialis brevis*, which attaches on a large area on the medial side of the crest, the *m. deltoideus clavicularis*, the attachment of which covers most of the lateral side of the crest, and the *m. supracoracoideus*, which attaches on the apex of the crest, whereas the *m.*

*pectoralis* only has a rather small attachment on the mediodistal side of the crest, just below the apex (*Meers, 2003*). This general arrangement seems to have been retained in sauropodomorphs (*Remes, 2008*; *Otero, 2018*) and basal theropods (*Burch, 2014*). In more derived theropods, the attachment site of the pectoralis muscle is enlarged, but not specifically marked on the medial side of the deltopectoral crest (*Jasinoski et al., 2006*), whereas the apex of the crest, potentially still serving for the attachment of the supracoracoides muscle, might be slightly expanded transversely, as in *Ornitholestes* (pers. obs. on AMNH 619 by CF and OR; *Figure 16A*) and the basal paravian *Mei* (IVPP V 12733). In most basal paravians (*Figure 16B*) and *Archaeopteryx*, the deltopectoral crest is large, but thin, and lacks an enlarged and marked attachment area for the pectoralis muscle (London, Maxberg and Thermopolis specimens; *Figure 13*). In contrast, later avialans, such as *Sapeornis* (*Figure 14*; *Provini et al., 2009*), *Jeholornis* (*Lefèvre et al., 2014*), *Jixiangornis* (*Chiappe and Meng, 2016*), *Confuciusornis* (e.g. JME 1997/1; *Chiappe et al., 1999*), Enantiornithes (*Walker et al., 2007*), *Ichthyornis* (*Clarke, 2004*), and many modern birds (*Figure 16D*) have a well-developed, anteromedially facing facet for the insertion of this most important flight muscle on the mediodistal part of the deltopectoral crest. *Alcmonavis* also shows such a marked facet on the medial side of the deltopectoral crest (*Figures 3* and *16C*), which is smaller than in many modern birds, but comparable in development to the facet seen in *Confuciusornis*, *Jeholornis*, *Jixiangornis* and *Sapeornis*. This indicates an increase in importance of the pectoralis muscle in avialan evolution after *Archaeopteryx*.

Another marked muscle attachment in *Alcmonavis* is the *tuberculum bicipitale radii* on the proximal radius. This tubercle is the attachment site of one of the branches of the *m. biceps brachii*, which extends from its origin on the coracoid (and anterior side of the proximal humerus in many groups) to insert on the proximal ends of radius and ulna in amniotes generally (*Remes, 2008*). In most dinosaurs, the insertion of the *m. biceps brachii* on the proximal radius is only noted by a rugose patch, but no marked tubercle is present. In contrast, birds usually have a well-developed tubercle on the anteromedial side of the proximal radius, as it is also present in *Alcmonavis* (*Figure 7A,B*) and other Mesozoic birds (e.g. *Confuciusornis*, SNSB-BSPG 1999 I 15, *Figure 15C*; *Sapeornis*, *Provini et al., 2009*; *Archaeorhynchus*, *Zhou et al., 2013*). Although this structure has not been described in

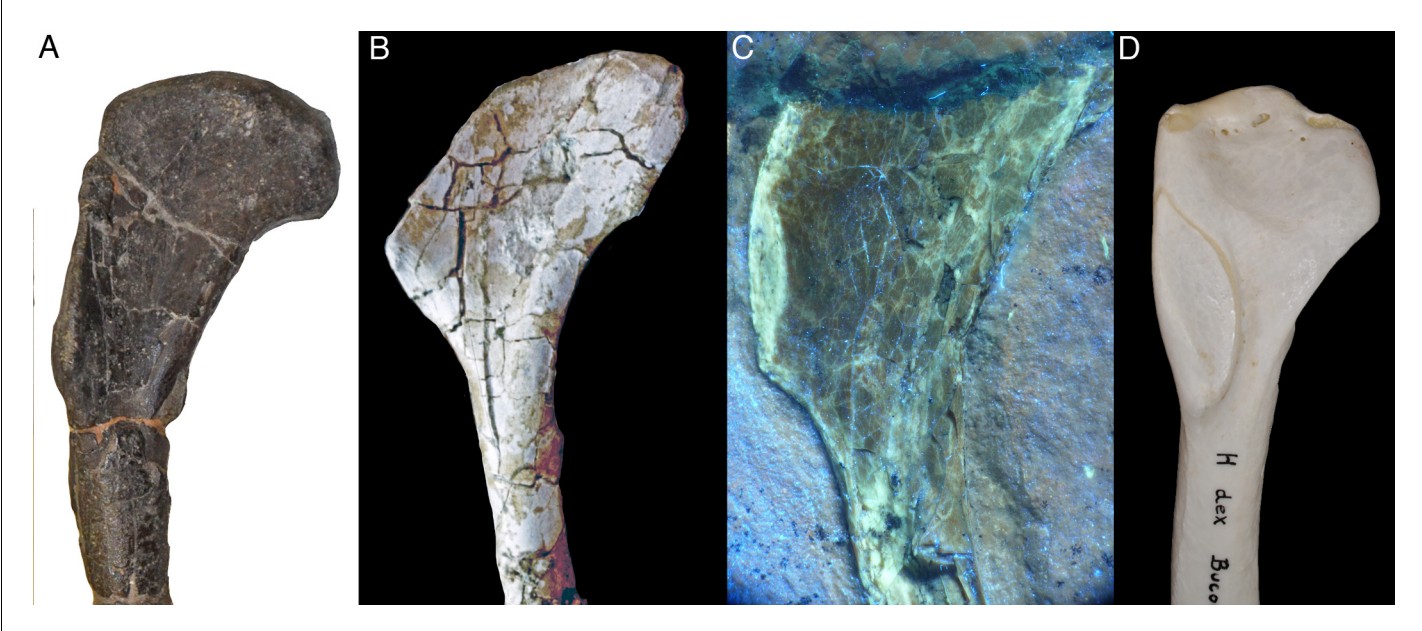

**Figure 16.** Development of the deltopectoral crest and the facet for the pectoralis muscle in several theropods. (A) Right humerus of the basal maniraptoriform *Ornitholestes hermani* (AMNH 619). (B) Left humerus (reversed for comparison) of the dromaeosaurid *Unenlagia comahuensis* (MCF PVPH 78). (C) Right humerus of *Alcmonavis poeschli* (under ultraviolet light; SNSB-BSPG 2017 I 133). (D) Right humerus of *Bucorvus abyssinicus* (northern ground hornbill; SNSB-BSPG unnumbered).
DOI: https://doi.org/10.7554/eLife.43789.023

*Archaeopteryx*, a small, triangular tubercle can actually be identified in several specimens, including the Munich (*Figure 17A*), Altmühltal and Ottmann and Steil (*Figure 17B*) specimens. We could not identify this structure in any specimen of *Anchiornis* we have seen or that is illustrated in the literature, nor in any other anchiornithid. Outside avialans, a marked tuberculum *bicipitale radii* is only present in *Microraptor* (IVPP V 13352) and, apparently in hypertrophied form, in *Bambiraptor* (*Burnham, 2004*). Thus, with the exception of *Bambiraptor*, a marked tubercle for the insertion of *m. biceps brachii* seems only to be present in volant forms.

Both the presence of a marked attachment area for the pectoralis muscle and the well-developed tubercle for the insertion of *m. biceps brachii* might thus have implications for the early evolution of flight. Whereas the role of the pectoralis muscle as the main downstroke muscle has been rather well studied, the function of other forelimb muscles, such as the *m. biceps brachii*, are less well understood (*Biewener, 2011*; *Tobalske, 2016*). As for the latter muscle, its primary function is usually considered to be the flexion of the forearm and the stabilization of the elbow joint, especially during the downstroke (*Dial, 1992*; *Biewener, 2011*; *Robertson and Biewener, 2012*). However, a recent study of muscle activity during flight in pigeons (*Robertson and Biewener, 2012*) also shows that this muscle has its highest activity patterns during take-off, indicating that it might be important for flapping take-off in modern birds in general.

The development of a pronounced tubercle for the insertion of the *m. biceps brachii* on the radius in volant basal avialans is thus consistent with the idea that these animals used a primitive form of flapping flight (*Carney, 2016*; *Heers and Carney, 2017*), probably starting as burst fliers, as recently suggested for *Archaeopteryx* by *Voeten et al., 2018*, based on cross-sectional geometry of the long bones of the forelimb. Thus, although a potential flight performance of the most basal avialans like *Anchiornis* is controversial (*Evangelista et al., 2014*; *Dececchi et al., 2016*; *Pan et al., 2019*), active flapping flight might have originated early within the lineage (see also *Meseguer et al., 2012*; *Dececchi et al., 2016*). If this primitive flapping flight was preceded by an intermediate gliding stage cannot be evaluated for the moment and requires more detailed studies on the ecology and life style of *Anchiornis* and its closest relatives. However, the appearance of a pronounced attachment site for the *m. pectoralis* and *m. biceps brachii* in *Alcmonavis* (in comparison with *Archaeopteryx*), a phylogenetically next step in the evolution towards modern birds, then possibly indicates an early improvement in flapping flight capabilities already in the Late Jurassic. This is also in accordance with the increased robusticity of metacarpal II and phalanx II-1, which form the *digitus majoris* in more derived birds, which serves as the attachment for the flight primaries.

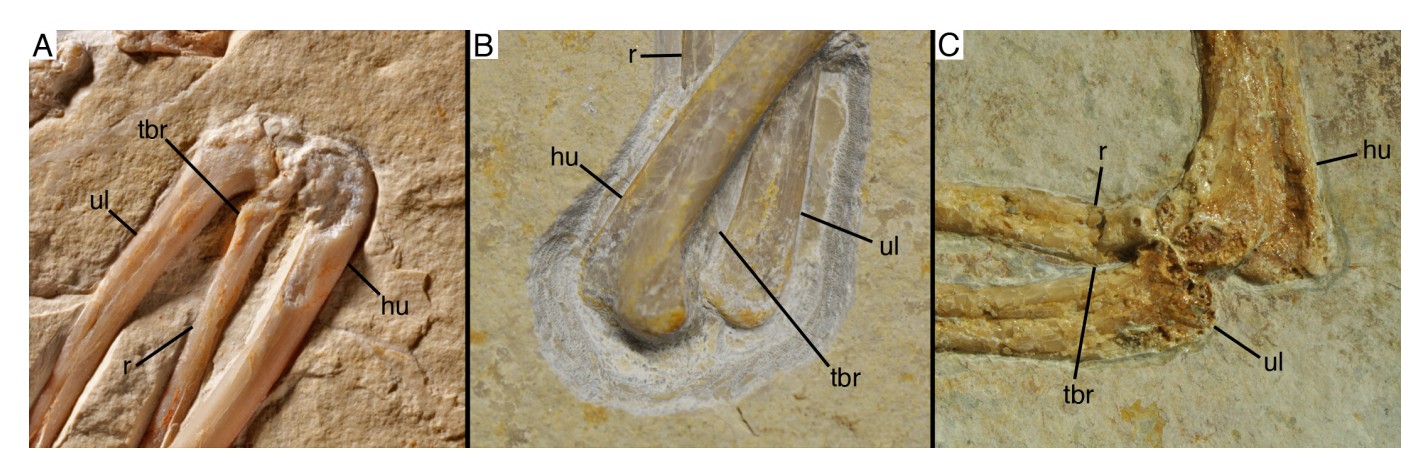

**Figure 17.** Development of the *tuberculum bicipitale radii* in basal avialan theropods. (**A**) Right elbow joint of *Archaeopteryx* (Munich specimen). (**B**) Right elbow joint of the Ottmann and Steil specimen (probably *Archaeopteryx*). (**C**) Left elbow joint of *Confuciusornis sanctus* (SNSB-BSPG 1999 I 15). Abbreviations: hu, humerus; r, radius; tbr, tuberculum bicipitale radii; ul, ulna.
DOI: https://doi.org/10.7554/eLife.43789.024

## Materials and methods

### Materials

The primary specimen described here, SNSB-BSPG 2017 I 133, was discovered in the old Schöpfel Quarry at the Schaudiberg, near Mühlheim, by R. Pöschl in 2017. The specimen was mechanically prepared by U. Leonhardt and subsequently purchased by the State of Bavaria, where it will be permanently stored at the Bayerische Staatssammlung für Paläontologie und Geologie in Munich. Numerous specimens of paravian theropods were studied for comparison, including all other urvogel specimens from the Solnhofen Archipelago (the currently missing Maxberg specimen could only be studied on the basis of a high quality cast at the BSPG) and several specimens of anchiornithids and basal avialans from China.

### Nomenclatural acts

The electronic edition of this article conforms to the requirements of the amended International Code of Zoological Nomenclature, and hence the new names contained herein are available under that Code from the electronic edition of this article. This published work and the nomenclatural acts it contains have been registered in ZooBank, the online registration system for the ICZN. The Zoo-Bank LSIDs (Life Science Identifiers) can be resolved and the associated information viewed through any standard web browser by appending the LSID to the prefix 'http://zoobank.org/". The LSID for this publication is: urn:lsid:zoobank.org:act:668F42B6-5BDC-4ADF-B271-36C6A43C7DB3. The electronic edition of this work was published in a journal with an ISSN.

### Anatomical nomenclature

In accordance with *Wilson, 2006*, we generally use the anatomical terms and orientation of bones as commonly used in the palaeontological literature on dinosaurs, rather than rigorously applying the skeletal terms proposed in the Nomina Anatomica Avium (*Baumel and Witmer, 1993*). However, the latter are used in respect to anatomical features that are typical for birds, but not present in non-avialan dinosaurs. Thus, we prefer the terms 'anterior' and 'posterior' over 'cranial' and 'caudal' (as the latter might be confused with anatomical regions of the skeleton), and refer to the different sides of long bones according to their orientation in the resting pose in a theropod dinosaur.

### UV imaging of *Alcmonavis poeschli*

Many fossils from the Upper Jurassic plattenkalks of southern Germany are fluorescent under artificial ultraviolet light (UV) which allows a more precise investigation of morphological details of skeletal remains as well as of soft parts. Since each fossil fluoresces slightly differently, a variety of filters and high performance UV-A lamps is required for investigation and imaging (*Tischlinger and Arratia, 2013*; *Tischlinger, 2015*). For our investigation of the 13th urvogel we used different UV-lamps with wavelengths of 312 nanometers (UV-B) and 365–366 nm (UV-A).

During the UV pictorial documentation of the 13th urvogel best results were obtained with a wavelength of 365–366 nanometers (long-wave radiation, UV-A). The following UV lamps were used:

3 Benda UV lamps: type N, 16 Watt, UV-A, 366 nanometers (size of filter 200 mm x 50 mm);

1 Labino UV lamp: UV-Spotlight S135, 35 Watt, UV-A, peak at 365 nm: spotlight (>50.000 microwatts per $cm^2$ at 30 cm distance) plus midlight reflector replacement (>8.000 microwatts per $cm^2$ at 30 cm distance).

The visibility of details under UV was enhanced considerably by an established filtering technique, crucial for the photographic documentation. The application of different filters allowed a selective visualisation of peculiar fine structures. Color compensation filters (yellow, cyan and magenta of different types and densities) were adjusted in front of the camera lens or under the microscope objective lens (if pictures were taken through the microscope). The optimum number and specification of the compensation filters was tested in a series of experiments. The predominant color of luminescence was of minor importance. In fact, the crucial decision on the amount of filtering was the optimal visibility of details and their differentiation from surrounding structures and the matrix (*Tischlinger and Arratia, 2013*).

## Phylogenetic analysis

In order to test the phylogenetic position of the new taxon, we included it in a modified version of the phylogenetic data matrix used in *Foth and Rauhut, 2017*. We furthermore checked codings for several taxa, added the recently described dromaeosaurid *Zhenyuanlong* (*Lü and Brusatte, 2015*), the troodontid *Jianianhuanlong* (*Xu et al., 2017*) and the avialan *Jinguofortis* (*Wang et al., 2018*) and four new characters. These characters are:

Character 562. Attachment for *m. pectoralis* on deltopectoral crest of humerus: not specifically marked, distal edge of deltopectoral crest might be slightly expanded (0); marked as an elongate oval, anteromedially inclined facet on the mediodistal surface of the deltopectoral crest (1). See discussion of this character above and *Figures 14* and *16*.

Character 563. Proximal articular surface of ulna: anteroposteriorly concave and flat or slightly convex transversely (0); developed as a round to oval concavity with slightly raised rims (1). In basal theropods and basal coelurosaurs, the ulna has a single proximal articular surface that is developed as an anteroposteriorly concave and transversely slightly convex to flat facet, in which the margins of the articular surface are not specifically marked (*Figure 18A*). Although the proximal end of the ulna is poorly exposed in the available specimens of Archaeopteryx, this also seems to be the condition in this taxon. In contrast, in birds, including *Alcmonavis* and other Mesozoic birds, such as *Confuciusornis*, there is a pronounced, round to oval concavity with slightly raised margins for the articulation with the ulnar condyle of the humerus (*Figure 18B*).

Character 564. Tuberculum bicipitale radii on the proximal radius: absent or indistinct (0); pronounced as a marked tubercle or crest (1). See discussion of this character above and *Figure 17*.

Character 565. Manual phalanx II-1: not significantly broadened when compared to other manual phalanges (0); strongly broadened, more than 1.5 times the width of phalanx II-2 and phalanges of digit III (1). See discussion of this character above and *Figure 12*.

The complete matrix had 136 taxa scored for 565 characters (see supplementary files). The matrix was analysed using the software TNT 1.5 (*Goloboff et al., 2008*; *Goloboff and Catalano, 2016*)

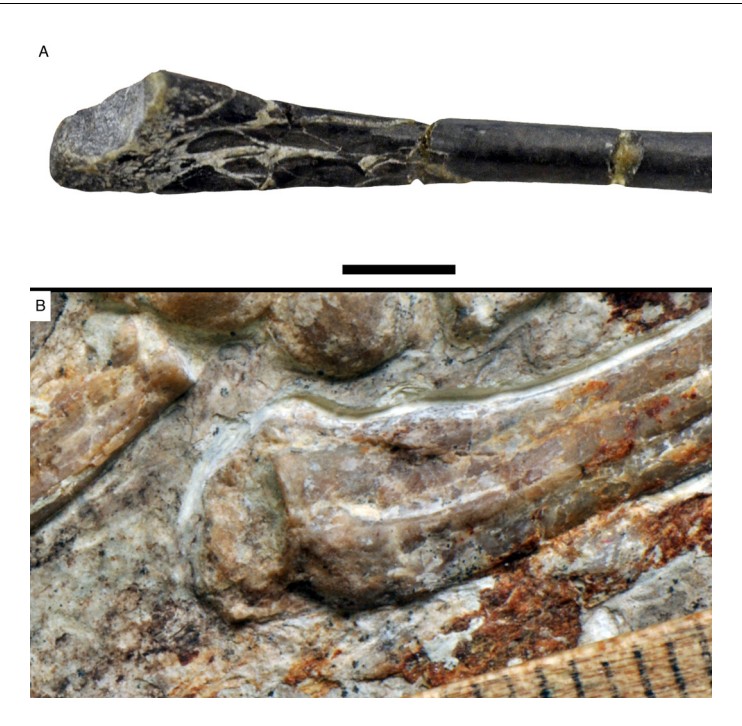

**Figure 18.** Proximal end of theropod ulnae, demonstrating character states for character 563. (**A**) Left ulna of Coelurus fragilis (YPM 2010). (**B**) Left ulna of Confuciusornis sanctus (JME 1997/1) Scale bar is 10 mm in (A) and in mm in (B).

DOI: https://doi.org/10.7554/eLife.43789.025

under equally weighted parsimony through a heuristic search of 10,000 replicates of Wagner trees followed by TBR branch swapping.

In order to further test the position of the new taxon, we ran a second analysis using implied weights (K = 12; *Goloboff et al., 2018*). This analysis was also carried out using TNT with 1000 replicates of Wagner trees followed by TBR branch swapping.

## Statistics

In order to statistically evaluate the significance of differences in proportions, we performed one sample parametric t-test (all samples show normal distribution) for the ratios of metacarpal III/metacarpal II, manual phalanx I-1/ulna, manual ungual I/ulna, manual phalanx II-1/manual phalanx I-1, manual phalanx II-1/manual ungual II, manual phalanx II-2/manual phalanx I-1, manual phalanx III-1/manual phalanx I-1, manual phalanx III-2/ulna, manual phalanx III-2/manual phalanx III-1, manual phalanx III-3/manual phalanx III-2, and manual digit I/manual digit II, testing the Mühlheim specimen against those specimens that can be classified as *Archaeopteryx* based on the diagnosis provided by *Rauhut et al., 2018*. Ratios including metacarpal I were not included due to uncertainties in the length of the bone. The one sample t-test compares the value in question with the range of the comparative statistical population of *Archaeopteryx* to evaluate the probability that this value represents the same population. The test was performed with help of the software PAST 3.21 (*Hammer et al., 2001*).

## Acknowledgements

Thanks are due to Roland Pöschl, who found the fossil, and Uli Leonhardt, who prepared the specimen and made it available for study. Martin Röper provided access to specimens and helped with discussions on localities within the Solnhofen Archipelago. Adriana López-Arbarello is thanked for helpful discussions, and Jingmai O'Connor, Trevor Worthy and two anonymous reviewers provided valuable comments that greatly helped to improve the paper. Numerous colleagues allowed access to paravian theropod specimens under their care; all this help is greatly appreciated. Judith Braukämper is thanked for additional technical support and the Generaldirektion der Staatlichen naturwissenschaftlichen Sammlungen Bayerns as well as the Bayerisches Staatsministerium für Wissenschaft und Kunst are acknowledged for their rapid and unbureaucratic action in providing the funds to secure the specimen for the BSPG. This work was supported by the Volkswagen Foundation under grant I/84 640 (to OR) and the Swiss National Science Fond under grant PZ00P2_174040 (to CF).

## Additional information

### Funding

| Funder | Grant reference number | Author |
| --- | --- | --- |
| Volkswagen Foundation | I/84 640 | Oliver WM Rauhut |
| Schweizerischer Nationalfonds zur Förderung der Wissenschaftlichen Forschung | PZ00P2_174040 | Christian Foth |

The funders had no role in study design, data collection and interpretation, or the decision to submit the work for publication.

### Author contributions

Oliver WM Rauhut, Conceptualization, Data curation, Software, Funding acquisition, Validation, Investigation, Visualization, Methodology, Writing—original draft, Writing—review and editing; Helmut Tischlinger, Resources, Visualization, Methodology, Writing—review and editing; Christian Foth, Formal analysis, Funding acquisition, Validation, Investigation, Methodology, Writing—original draft, Writing—review and editing

## Author ORCIDs

Oliver WM Rauhut iD http://orcid.org/0000-0003-3958-603X

## Decision letter and Author response

Decision letter https://doi.org/10.7554/eLife.43789.032
Author response https://doi.org/10.7554/eLife.43789.033

## Additional files

### Supplementary files

• Supplementary file 1. List of characters used in phylogenetic analysis and phylogenetic data matrix.
DOI: https://doi.org/10.7554/eLife.43789.026

• Supplementary file 2. Data matrix.
DOI: https://doi.org/10.7554/eLife.43789.027

• Transparent reporting form
DOI: https://doi.org/10.7554/eLife.43789.028

### Data availability

All data generated or analysed during this study are included in the manuscript and supporting files.

The following previously published dataset was used:

| Author(s) | Year | Dataset title | Dataset URL | Database and Identifier |
|---|---|---|---|---|
| Foth C, Rauhut OWM | 2017 | Re-evaluation of the Haarlem Archaeopteryx and the radiation of maniraptoran theropod dinosaurs | https://morphobank.org/index.php/Projects/ProjectOverview/project_id/2532 | MorphoBank, 2532 |

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
