## [Decision Letter]

Thank you for submitting your article "A non-archaeopterygid bird (Dinosauria; Theropoda; Avialiae) from the Late Jurassic of southern Germany" for consideration by *eLife*. Your article has been reviewed by four peer reviewers, including Trevor Worthy as Guest Editor and Reviewer #2, and the evaluation has been overseen by Diethard Tautz as the Senior Editor. The following individuals involved in review of your submission have agreed to reveal their identity: Jingmai Oconnor (Reviewer #4).

The reviewers have discussed the reviews with one another and the Reviewing Editor has drafted this decision to help you prepare a revised submission.

Summary:

This manuscript describes a new fossil, the thirteenth urvogel from the Solnhofen archipelago (where *Archaeopteryx* derives from), showing that it represents a second avialan taxon that is slightly more derived than *Archaeopteryx*. The conclusion that follows is that some flight adaptations and the transition towards Aves evolved earlier than previously thought. This is a major discovery - if substantiated - and thus one that requires extraordinary justification. You provide an excellent morphological description of an important new specimen, in a well written manuscript supplied with wonderful photographs in both normal and UV light. However, while it reveals some new anatomical details, the specimen is fragmentary, with only wing bones, and some of the key conclusions of the manuscript are rather poorly supported by the actual data as presented. In addition, some speculative ideas are presented as too definitive in the abstract and conclusion. Added comparisons and attention to the points raised below may possibly overcome these obstacles; hence a revision is invited.

Essential revisions:

1) Adequate Diagnosis from *Archaeopteryx*

Two reviewers have strong reservations as to whether the fossil is adequately diagnosed as distinct from *Archaeopteryx* and more ancestral avialan. There is also concern how well the erection of a new genus name for the fossil is substantiated. In the diagnosis, the authors differentiate the fossil from "all other theropods", but what would be needed is an explicit differentiation from *Archaeopteryx* and the newly erected "*Ostromia*".

Does the phrase "all theropods" in the diagnosis include birds as well? If it does, some of the features, such as a "large deltopectoral crest", are erroneously listed, as these do occur in avians (or "avialans" sensu the authors). One reviewer was not convinced by the diagnosis distinguishing it from other theropods, noting that the phalanges seem very mobile (whereas the digits are typically preserved straight indicating more rigidity (which would be necessary for flight) in *Archaeopteryx*; also the morphology of the semilunate is more non-avian than avian, and the same seems possible with regards to the distal humerus, and the proportions of the claws relative to their digits (not including horny sheath). We think a stronger diagnosis is therefore required. Throughout the description, anywhere comparison is only made with avians, non-avians should be added (and vice versa).

2) Comments on specific morphologies

There are reservations concerning the interpretation of the humerus that need addressing.

Some of the alleged differences between the new fossil and previously described specimens are controversial. For example, one of the key features listed by the authors, the alleged presence of an impression for musculus pectoralis, cannot be established with certainty in some other *Archaeopteryx* fossils, such as the London specimen, in which the deltopectoral crest appears to be abraded in the corresponding region (see Figure 12 of the manuscript). This facet in the new fossil does not appear to be substantially different from that found in Mei long (Figure 14B), and certainly it is much less developed than the corresponding muscular impression in more crownward birds. Reviewer Oconnor noted that the attachment for the pectoralis on the distal end of the deltopectoral crest has not been widely described, at least not in non-ornithothoracine Early Cretaceous birds (it is mentioned in Martinavis; Walker et al., 2007), so taxa that have this feature should be listed together with citations (be careful, although it may appear present from a figure, if it wasn't described, its appearance in a photograph is possibly an artefact of preservation/photography, e.g., *Sapeornis* IVPP V13396 – Provini et al., 2008, or the holotype of *Jeholornis*. It is not mentioned in any of the papers cited, and in Oconnor's experience with basal Jehol birds is not common and so must have been independently evolved in Alcomavis, or perhaps only appears in the most ontogenetically mature specimens of *Archaeopteryx* (and potentially other Mesozoic birds).

It is not accurate to estimate the length of the humerus as it is incomplete and to do so based on assumptions about the size of the internal tuberosity. Better to just state its minimal length (and not use this element for ratios and comparison). Which other taxa show elongate oval facet for the insertion of the m. pectoralis (make sure to check non-avians)?

With regards to the angle of the proximal portion of the humerus relative to the shaft the authors say to a lesser degree in non-avians but this seems a bit misleading – it is also strongly angled in a number of non-avian paravians, especially it seems in dromaeosaurids (Ji et al., 2000 – *Sinornithosaurus; Microraptor*, any number of references) but also present in *Anchiornis* (this is another reason why line drawing reconstructions of the forelimb in 4-5 key taxa would be really helpful). Reviewer Worthy suggested that the shaft angle (Subsection “Humerus”, etc) is perhaps increased by inclusion of the bicipital crest. The bicipital crest can be seen to join the shaft roughly co-level with the distal end of the deltopectoral crest, and thus the bicipital crest adds width to the ventral (here posteromedial) side of the shaft. Exclusion of the bicipital crest would lead to a lesser angle of the proximal section of the shaft relative to that more distally. Thus, using the ventral shaft margin as a proxy for shaft angle is not homologous between taxa that have a definite and enlarged bicipital crest and those that do not. A revision of definition of this metric would avoid this issue. Applies to the Discussion section as well.

Ulna – since the humerus is not fully preserved, it is not accurate to compare an estimated length and then derive comparisons. Compare proximal end of ulna to Anchiornis. Although caution is mentioned often, the distal ulna seems pretty crushed flat but still described heavily. The features described are only present in *Archaeorhynchus* or also present in other stem birds? Label the anteromedially directed ridge in Figure 6.

Radius – compare longitudinal groove to that in enantiornithines.

Carpals – MC I seems to be broken and it was unlikely it was as short as described. MC II is thicker than I and III in *Microraptor* as well; also, Mc III is slightly offset proximally.

Digits – Digit I ungual largest in *Anchiornis* (see Pei et al., 2017); digit II phalanx 1 is certainly robust but it appears strongly exaggerated as preserved, as two surfaces are exposed and flattened so it appears to be a single exposed surface of a hyper-robust phalanx. If this is taken into account, then the proximal articular surface of the phalanx is slightly narrower than the matching articular surface on metacarpal II. I do not see this described convexity on the plantar surface of phalanx II-2. The flexed morphology described in the penultimate phalanges of digits I and II is very subtle, at best, and should be described as so.

Unguals – it is stated (Discussion section) that the unguals are larger (proportionately) in *Archaeopteryx* but if the ungual (not including horny sheaths which may or may not be complete or in situ) is compared to the length of the penultimate phalanx, they are proportionately larger in Alcomavis. A table of ratios would be helpful to further help illustrate differences between this specimen and other urvogels and closely related taxa. *Confuciusornis* does not show a reduction in manual claws from I to III (II is smallest). Since this specimen is larger than other *Archaeopteryx* specimens, the pronounced muscle attachments could be explained by ontogeny (as suggested by the authors themselves somewhat), especially given that the difference between these taxa is pretty minor if the robustness of phalanx II-1 is taken out.

3) Advisability of creating a new nomen

Even if the new specimen can be shown to differ from other *Archaeopteryx* fossils in some features, it is contentious, and some would say not advisable to assign a name to these fragmentary remains. The past years have seen an inflation in new and rather poorly diagnosed *Archaeopteryx*-like avians from the Solnhofen limestone, and in light of the very fragmentary preservation of the fossil, the manuscript would be scientifically more rigorous if the authors opted against naming a new taxon.

4) Phylogenetic analysis

The phylogenetic analysis is presented in rather a cursory way. It is unacceptable not to have support values in the presented phylogenetic trees. For example, it is misrepresentative to say the strict consensus is well resolved (subsection “Phylogenetic position of *Alcmonavis”*) yet not show the evidence that there is in fact very weak support for such resolution. At the very least Bremner support values should be given so that readers can see which of these clades has any semblance of support – one assumes that as no values are given and given the statement, that clade support is very low indeed – but readers deserve to be shown this, not have these data occluded. Ideally bootstrap values over 50% should be shown because anything less shows the clades are highly unstable and probably are not real in terms of evolutionary history. As presented, the placement of *Alcmonavis* is discussed as an afterthought in this section. Surely the focus should be on the new taxon and then use known taxa secondly to validate the tree.

Given it is well established that missing data drive a taxon to the base of the clade in which it would otherwise reside (e.g., Sansom, 2015), some comment on the placement of the new taxon is warranted; why, what character evidence precludes that, for example, the new taxon really lies within Jelholornithidae or another slightly more derived position, than the node it appears in the tree? Character evidence for the critical nodes should be discussed, identifying the derived characters that place the new fossil in a more crownward position than *Archaeopteryx*.

Sansom, 2015.

The phylogenetic analyses should state which multistate characters were ordered (if any).

Major clade names, including all Family names used in text, should be shown on the trees in the Supplementary file.

5) Improvement of figures

While the currently presented figures are excellent, some more are needed. Key to demonstrating the distinction from *Archaeopteryx*, are photos and or line drawings of the important elements of the new fossil in direct comparison with those of "*Archaeopteryx*" that explicitly show the differences in morphology and proportions. It is also desirable to add some figures, at least in Supplementary file, of key features of the new fossil directly compared with a meaningful selection of stem birds and non-avian maniraptorans to aid with comparison.

6) Tabular presentation of key data to facilitate comparison

Use of two more tables might facilitate comparisons better. For example, the section on Comments on the numbering and naming of avialan specimens from the Solnhofen Archipelago is useful, but quite long. Perhaps include the data in a table listing all 13 specimens, their current taxonomy, their numbers and correlative 'names' and institution, and include just a couple sentences to justify the new names for the eleventh to thirteenth specimens. If length is an issue then it could be in Supplementary file, but it is of more use to the audience in the main text, i.e. not lost/invisible in the Supplementary file.

Similarly, a table with comparative key measurements for the 12 *Archaeopteryx* specimens including ratios should also be provided, so demonstrating how similar this specimen is to other urvogel specimens.

7) Discussion on possible flight mode of the fossil

This section is entirely speculative and not well substantiated. Various authors have detailed that flapping flight capabilities are unlikely for *Archaeopteryx*. The different morphology of the supracoracoideus muscle is just one argument contra flapping flight capabilities of *Archaeopteryx*, and the anatomy of *Archaeopteryx* does not show any evidence that this animal was capable of take-off through flapping flight (comparisons with extant pigeons certainly are not appropriate, as the latter have a completely different skeletal morphology than any Jurassic avians). Even though Voeten et al. recently proposed flapping flight capabilities of *Archaeopteryx*, their arguments based on the thickness of the bone cortex are far from compelling. Likewise, it is not clear why the presence of a tubercle for musculus *biceps brachii* on the radius would indicate flapping flight capabilities (so much the more as the presence/absence of this feature in other *Archaeopteryx* specimens is difficult to assess). Given the speculative nature of this interpretation, to include in the abstract a sentence like "Several modifications, especially in muscle attachments related to the downstroke of the wing, indicate a rapid adaptation of the forelimb for active flapping flight in the early evolution of birds" is not justified by the data (so much the more as the discussion is far less definitive).

Given the great similarities of the general skeletal architecture of this fragmentary wing to the known *Archaeopteryx* remains (three unfused metacarpals with large ungual phalanges, similar phalangeal proportions etc.), it is highly unlikely that substantial differences in flight mode to *Archaeopteryx* existed. The speculative nature of this discussion and interpretation should be addressed.

8) Biogeographic interpretations.

These speculations towards the end of the manuscript should be deleted altogether.

---

## [Author Response]

Essential revisions:1) Adequate Diagnosis from ArchaeopteryxTwo reviewers have strong reservations as to whether the fossil is adequately diagnosed as distinct from Archaeopteryx and more ancestral avialan. There is also concern how well the erection of a new genus name for the fossil is substantiated. In the diagnosis, the authors differentiate the fossil from "all other theropods", but what would be needed is an explicit differentiation from Archaeopteryx and the newly erected "Ostromia".

The explicit differentiation of *Alcmonavis* to *Archaeopteryx* and *Ostromia* was already provided in the first part of the Discussion section of the previous manuscript. To avoid confusion, we rearranged the manuscript and put the diagnosis after the description and discussion of the differences so that the reader already has this information before the new taxon is defined.

We extended the comparison between *Alcmonavis* to *Archaeopteryx* and *Ostromia* by including statistical analyses on various limb proportions.

Does the phrase "all theropods" in the diagnosis include birds as well? If it does, some of the features, such as a "large deltopectoral crest", are erroneously listed, as these do occur in avians (or "avialans" sensu the authors). One reviewer was not convinced by the diagnosis distinguishing it from other theropods, noting that the phalanges seem very mobile (whereas the digits are typically preserved straight indicating more rigidity (which would be necessary for flight) in Archaeopteryx; also the morphology of the semilunate is more non-avian than avian, and the same seems possible with regards to the distal humerus, and the proportions of the claws relative to their digits (not including horny sheath). We think a stronger diagnosis is therefore required. Throughout the description, anywhere comparison is only made with avians, non-avians should be added (and vice versa).

As stated in the text, *Alcomonavis* is diagnosed based on a combination of characters, i.e., single characters listed in the diagnosis do not have to be unique. Of course, large deltopectoral crests are typical for many basal birds, but in its extension the deltopectoral crest of *Alcmonavis* is larger than in *Archaeopteryx*, anchiornithids and other non-avian theropod dinosaurs.

The reviewer is correct that in many pygostylian birds, the digits are rigid, especially in Ornithothoraces. This also seems to be the case in most *Archaopteryx* specimens, but also in *Anchiornis* or *Microraptor*. However, we believe that the preservation of the phalanges in *Alcmonavis* is not an indicator of missing rigidity, but a taphonomic artefact caused by an advanced stage of decay; indeed, fossils of the Mörnsheim Formation tend to be more strongly disarticulated and disrupted than those of the underlying Altmühltal Formation, which has yielded the vast majority of specimens of *Archaeopteryx*. Moreover, some specimens of *Archaeopteryx*, including the "chicken wings" and the twelfth specimen, also show strongly flexed phalanges. Indeed, similar “mobile” preservations can be also found in *Sapeornis* (IVPP V13396), *Pengornis* (IVPP V15336), *Gansus* (BMNHC-PH1392) and many specimens of *Confuciusornis* (DNHM-D2454, IVPP V10928, IVPP V12352, IVPP V13156 or JME 1997-1) (see e.g., Falk et al., 2016; Wang et al., 2019 or Chiappe and Meng, 2016), so we consider this simply to be an artefact of preservation.

We already stated that the morphology of the semilunate carpal of *Alcomonavis* resembles that of non-avian theropods like *Deinonychus*, being not fused to the metacarpus. However, the latter situation is also not present in other basal birds, including *Jeholornis, Sapeornis* or *Confuciusornis*. In fact, a fusion of carpal and metacarpal bones happens first time within Ornithothoraces, like in *Archaeorhynchus* (see e.g., Botehlo et al., 2014). However, even this partial fusion in *Archaeorhynchus* seem to be occur very late in ontogeny (Wang and Zhou, 2017). Despite the fact that we did not include any diagnostic character from the carpal region, we added a comparison to the description.

All measurements of the unguals based on the bony element. The sheath was not taken into account. Our statistical analyses shows that UI is significant smaller when compared to the ulna length and UII when compared to the length of manual phalanx II-1.

Concerning the diagnosis we don't see how the "strength" of the diagnosis depends on how bird-like or not the new taxon is – although no clear autapomorphic characters can currently be identified, the combination of characters seen in this wing skeleton is unique within both non-avialan and avialan theropods.

We have added comparisons with derived maniraptoran theropod dinosaurs.

2) Comments on specific morphologiesThere are reservations concerning the interpretation of the humerus that need addressing.Some of the alleged differences between the new fossil and previously described specimens are controversial. For example, one of the key features listed by the authors, the alleged presence of an impression for musculus pectoralis, cannot be established with certainty in some other Archaeopteryx fossils, such as the London specimen, in which the deltopectoral crest appears to be abraded in the corresponding region (see Figure 12 of the manuscript). This facet in the new fossil does not appear to be substantially different from that found in Mei long (Figure 14B), and certainly it is much less developed than the corresponding muscular impression in more crownward birds. Reviewer Oconnor noted that the attachment for the pectoralis on the distal end of the deltopectoral crest has not been widely described, at least not in non-ornithothoracine Early Cretaceous birds (it is mentioned in Martinavis; Walker et al., 2007), so taxa that have this feature should be listed together with citations (be careful, although it may appear present from a figure, if it wasn't described, its appearance in a photograph is possibly an artefact of preservation/photography, e.g., Sapeornis IVPP V13396 – Provini et al., 2008, or the holotype of Jeholornis. It is not mentioned in any of the papers cited, and in Oconnor's experience with basal Jehol birds is not common and so must have been independently evolved in Alcomavis, or perhaps only appears in the most ontogenetically mature specimens of Archaeopteryx (and potentially other Mesozoic birds).

The reviewer is right that the margin of the deltopectoral crest in London specimens is slightly abraded, increasing towards proximal. More distal the margin of the crest is still intact so that the presence of a facet can be ruled out (pers. obs. by OR). Furthermore, the crest is partially preserved also in the Thermopolis and Maxberg specimen (cast of the latter at the BSPG), none of which shows any sign of such a facet. To strengthen our case, we added a photo of the right humerus of the Thermopolis specimen, which shows the same situation to Figure 12. Furthermore, we added another photo showing the same region in *Alcmonavis* for comparison.

Based on the figure we provided in the previous version, it appears that *Mei long* possesses such a facet, but actually the photo shows the margin of the crest, which faces anterior (in many theropods the deltopectoral crest is not flat but slightly curved). Like in *Ornitholestes* (discussed in the manuscript) the margin is slightly expanded transversally. We changed the relevant sections in the manuscript accordingly.

We do not follow the argumentation that the appearance of undescribed structures in photographs are artefacts of preservation/photography. We believe that the illustration of anatomical details is part of a description, and it becomes even more essential, when the written part is rather brief (which is often the case, unfortunately). In this context, the documentation of preservational artefacts is also necessary to avoid misinterpretations. Nevertheless, the presence of the facet in *Sapeornis* is confirmed by pers. obs. of CF, and in *Confuciusornis* by pers. obs. of OR. The structure of the respective specimen is now illustrated in Figure 14. As it is not properly figured, we did not cite the holotype of *Jeholornis*, but a referred specimen, which was described by Lefèvre et al., (2014).

Although muscle attachments often become more conspicuous during ontogeny, the London specimen, which, as shown, does not have such as facet, is certainly not a young individual, and is not so much smaller than the new specimen. Thus, this specimen should show some sort of pronounced facet, which is not the case.

It is not accurate to estimate the length of the humerus as it is incomplete and to do so based on assumptions about the size of the internal tuberosity. Better to just state its minimal length (and not use this element for ratios and comparison). Which other taxa show elongate oval facet for the insertion of the m. pectoralis (make sure to check non-avians)?

We now added the actual length of the humerus as preserved. However, as fossils often suffered damage during the preservation or excavation process it is common in palaeontology to provide estimated length for respective elements. The estimation of the humerus length is based on ratios of complete elements known from other paravians. Nevertheless, as the correct length of the element is not known, the humerus length was not considered for statistical comparisons.

In the previous version, taxa with an oval facet were listed in the Discussion section. For comparison, we added taxa to the description and made a reference to the discussion.

With regards to the angle of the proximal portion of the humerus relative to the shaft the authors say to a lesser degree in non-avians but this seems a bit misleading – it is also strongly angled in a number of non-avian paravians, especially it seems in dromaeosaurids (Ji et al., 2000 – Sinornithosaurus; Microraptor, any number of references) but also present in Anchiornis (this is another reason why line drawing reconstructions of the forelimb in 4-5 key taxa would be really helpful). Reviewer Worthy suggested that the shaft angle (Subsection “Humerus”, etc) is perhaps increased by inclusion of the bicipital crest. The bicipital crest can be seen to join the shaft roughly co-level with the distal end of the deltopectoral crest, and thus the bicipital crest adds width to the ventral (here posteromedial) side of the shaft. Exclusion of the bicipital crest would lead to a lesser angle of the proximal section of the shaft relative to that more distally. Thus, using the ventral shaft margin as a proxy for shaft angle is not homologous between taxa that have a definite and enlarged bicipital crest and those that do not. A revision of definition of this metric would avoid this issue. Applies to the Discussion section as well.

The reviewer is correct that the proximal ends of the humeri of *Anchiornis, Sinornithosaurus* and *Microraptor* are angled, but less than 30 percent. We already provided these values for Dromaeosauridae and *Anchiornis* in the Discussion section of the previous version. We added this comparison also to the description.

The bicipital crest (internal tuberosity in non-avialan theropods) is usually included in this measurement, so the values should be comparable.

Ulna – since the humerus is not fully preserved, it is not accurate to compare an estimated length and then derive comparisons. Compare proximal end of ulna to Anchiornis. Although caution is mentioned often, the distal ulna seems pretty crushed flat but still described heavily. The features described are only present in Archaeorhynchus or also present in other stem birds? Label the anteromedially directed ridge in Figure 6.

The most detailed description of *Anchiornis* (including photos) was provided by Pei et al., 2017. Unfortunately, they describe only a minor expansion and a reduced olecranon process, which is a common morphology in Maniraptora. Nothing is stated concerning the cotylae. Neither the figures in Pei et al., 2017 nor personal photos of various *Anchiornis* specimens taken by CF can help to add more comparison.

The distal expansion of the ulna is more common in Avialae, so we added more comparisons. It is also present in *Archaeopteryx*, but is less pronounced and more symmetrically shaped.

We have only described features of the distal ulna that can be established despite the crushing of the bone.

The ridge was labelled in the figure.

Radius – compare longitudinal groove to that in enantiornithines.

We added a comparison into the description section and also discussed the presence of this character in the Discussion section.

Carpals – MC I seems to be broken and it was unlikely it was as short as described. MC II is thicker than I and III in Microraptor as well; also, Mc III is slightly offset proximally.

We noted in the description that Mc I is incomplete. How short this element was is everybody's guess; we only say that IF the position of the first phalanx indicates the distal end of this metacarpal (which is the case in several other similarly preserved theropod specimens), this element was very short. However, we did not use the possible length of this metacarpal for any formal analysis.

Unfortunately, we cannot evaluate this based on the scientific literature. Personal observations of several specimens by CF shows that the proximal end of MCI is strongly expanded in *Microraptor* (see also measurements provided by Hwang et al., 2002), while it becomes thinner distally. However, this is not the exact situation we found in *Alcmonavis* and *Jeholornis*. Generally, Mc II seems to be less robust in *Microraptor* than in *Alcmonavis*.

Because the metacarpus of *Alcmonavis* shows no signs of disarticulation, the offset position of Mc III most likely represents the actual situation. As already stated in the text, this morphology is present in many other Avialae, but also in several oviraptorids and dromaeosaurids (these comparisons were added to the text).

Digits – Digit I ungual largest in Anchiornis (see Pei et al., 2017);

The measurements provided for *Anchiornis* by Hu et al., 2009; Pei et al., 2017 and Guo et al., 2018 all show that the ungual of digit II is slightly longer than that of digit I. Personal observation of the holotype by CF confirms this.

digit II phalanx 1 is certainly robust but it appears strongly exaggerated as preserved, as two surfaces are exposed and flattened so it appears to be a single exposed surface of a hyper-robust phalanx. If this is taken into account, then the proximal articular surface of the phalanx is slightly narrower than the matching articular surface on metacarpal II. I do not see this described convexity on the plantar surface of phalanx II-2. The flexed morphology described in the penultimate phalanges of digits I and II is very subtle, at best, and should be described as so.

We gave this character a lot of thought and scrutiny of the actual fossil. Although the phalanx might give the impression of two surfaces being exposed, this does not seem to be the case; especially at the proximal end we are confident that this is the true width of the phalanx, whereas the distal end seems to show some internal twist in the phalanx, giving the impression that both palmar and lateral surface are exposed. Again, however, detailed investigation of the specimen indicate that this is not due to compression, but largely reflects the true morphology (although some compression is evident, as stated in the description), with the distal articular surface being slightly twisted. We added a sentence on the preservation in the description.

Concerning the flexed morphology of the penultimate phalanges, we noted in the description that these elements are "slightly bowed" or "slightly flexed". Furthermore, their morphology can be seen in the figure.

Unguals – it is stated (Discussion section) that the unguals are larger (proportionately) in Archaeopteryx but if the ungual (not including horny sheaths which may or may not be complete or in situ) is compared to the length of the penultimate phalanx, they are proportionately larger in Alcomavis. A table of ratios would be helpful to further help illustrate differences between this specimen and other urvogels and closely related taxa. Confuciusornis does not show a reduction in manual claws from I to III (II is smallest). Since this specimen is larger than other Archaeopteryx specimens, the pronounced muscle attachments could be explained by ontogeny (as suggested by the authors themselves somewhat), especially given that the difference between these taxa is pretty minor if the robustness of phalanx II-1 is taken out.

Statistical comparisons with *Archaeopteryx* reveal that UI of the Altmühl specimen is relatively smaller compared to the Ulna, while UII is relatively smaller compared to the preungular phalanx, both on a significant level. In contrast, UI is not relatively smaller compared to PI-1, because the latter element is relatively shortened, too.

We corrected the observation concerning the claw reduction in *Confuciusornis*.

As morphology and relative size of the relevant muscle attachments sites on humerus and radius do not really vary throughout the different specimens of *Archaeopteryx*, ontogenetic variation would imply drastic morphological changes in the final phase of ontogeny. However, as growth curves of tetrapods usually have a logarithmic, inverted exponential or logistic shape (Karkach, 2006), significant shape changes are rather expected in the phase of early ontogeny. Therefore, we consider the differences as interspecific.

3) Advisability of creating a new nomenEven if the new specimen can be shown to differ from other Archaeopteryx fossils in some features, it is contentious, and some would say not advisable to assign a name to these fragmentary remains. The past years have seen an inflation in new and rather poorly diagnosed Archaeopteryx-like avians from the Solnhofen limestone, and in light of the very fragmentary preservation of the fossil, the manuscript would be scientifically more rigorous if the authors opted against naming a new taxon.

We disagree with the reviewer that there is "an inflation in new and rather poorly diagnosed *Archaeopteryx*-like avians from the Solnhofen limestone" – on the contrary, there rather is a problem with the assumption that all avialian theropods from the Solnhofen Archipelago (please see the manuscript for why we prefer not to talk about the "Solnhofen limestone") should be *Archaeopteryx*, as we now know that other, bird-like theropods were around in the Late Jurassic, as outlined by Rauhut et al., (2018) and in the current manuscript. Thus, in an attempt to better diagnose the genus *Archaeopteryx*, we realized that some specimens cannot be referred to this genus with certainty, and that one specimen, in particular (the Haarlem specimen), showed significant differences. As a species name was already available for this specimen, we thus proposed a new genus name, *Ostromia* (Foth and Rauhut 2017). However, this is the only non-archaeopterygid paravian theropod named from the Solnhofen Archipelago (and, as outlined, we only provided a new generic name for an existing species that can be shown to not fit in the genus *Archaeopteryx*), so this is hardly "an inflation in new and rather poorly diagnosed *Archaeopteryx*-like avians".

Regarding the new specimen, we approached its study with the same premise: First we tried to clarify if the specimen can be referred to *Archaeopteryx* with any certainty. Apart from the general proportions – which, as outlined in the manuscript, do also not differ significantly from several other paravians and basal avialans – there is no reason to refer this specimen to the genus *Archaeopteryx*. As our detailed study of the specimen revealed several differences to all other specimens of *Archaeopteryx*, the character combination shown by this specimen is unique in theropods in general, and a phylogenetic analysis found this specimen in an intermediate position between *Archaeopteryx* and later avialans, we opted to create a new taxon, after considering alternative options for almost a year.

Fragmentary preservation is not in itself an argument against naming new taxa, as demonstrated by literally thousands of fossil vertebrate species (especially in palaeoornithology it is not unusual that new species are based even on single bones) – what matters is if a new taxon can be adequately diagnosed. As the character combination demonstrated by the new specimen is unique within theropods (as also supported by the phylogenetic analysis), we argue that this is the case for *Alcmonavis*.

4) Phylogenetic analysisThe phylogenetic analysis is presented in rather a cursory way. It is unacceptable not to have support values in the presented phylogenetic trees. For example, it is misrepresentative to say the strict consensus is well resolved (subsection “Phylogenetic position of Alcmonavis”) yet not show the evidence that there is in fact very weak support for such resolution. At the very least Bremner support values should be given so that readers can see which of these clades has any semblance of support – one assumes that as no values are given and given the statement, that clade support is very low indeed – but readers deserve to be shown this, not have these data occluded. Ideally bootstrap values over 50% should be shown because anything less shows the clades are highly unstable and probably are not real in terms of evolutionary history. As presented, the placement of Alcmonavis is discussed as an afterthought in this section. Surely the focus should be on the new taxon and then use known taxa secondly to validate the tree.

We estimated bootstrap and Bremer support values for each node. Bootstrap values are generally low. For the equal-weight parsimony analysis, the phylogenetic position of *Alcmonavis* is supported by a Bremer support of 2, which is higher than for most other clades, including the phylogenetic position of *Archaeopteryx* (implied-weight parsimony leads to similar results). The support values for each node are shown for the strict consensus trees in the Supplementary file. The updated version of Figure 15. shows now the Bremer support values for the Avialae.

Given it is well established that missing data drive a taxon to the base of the clade in which it would otherwise reside (e.g., Sansom, 2015), some comment on the placement of the new taxon is warranted; why, what character evidence precludes that, for example, the new taxon really lies within Jelholornithidae or another slightly more derived position, than the node it appears in the tree? Character evidence for the critical nodes should be discussed, identifying the derived characters that place the new fossil in a more crownward position than Archaeopteryx.

In the Discussion section of the phylogeny, we listed the characters that support the more crownward position than *Archaeopteryx*.

We agree that it could be possible that the missing data places *Alcmonavis* closer to the base of Avialae. As the autapomorphies for the more derived position of *Jeholornis, Sapeornis* and co. are based mainly on forelimb characters, we believe that the phylogenetic position of *Alcmonavis* is robust.

Sansom, 2015.The phylogenetic analyses should state which multistate characters were ordered (if any).

We provided a character list in the previous version. Ordered characters were labelled as Ordered.

Major clade names, including all Family names used in text, should be shown on the trees in the Supplementary file.

For the two reduced consensus trees we added the most important taxa names that are also mentioned in the text at the relevant nodes.

5) Improvement of figuresWhile the currently presented figures are excellent, some more are needed. Key to demonstrating the distinction from Archaeopteryx, are photos and or line drawings of the important elements of the new fossil in direct comparison with those of "Archaeopteryx" that explicitly show the differences in morphology and proportions. It is also desirable to add some figures, at least in Supplementary file, of key features of the new fossil directly compared with a meaningful selection of stem birds and non-avian maniraptorans to aid with comparison.

We added a figure comparing the outlines of humeri of several avialan theropods for comparison, plus a figure showing an actual specimen of *Sapeornis* in the taxonomic Discussion section. We furthermore added several basal paravian and basal avialan mani in the figure showing the manus of *Archaeopteryx* and changed part of the figure comparing the attachment area of the pectoralis muscle.

6) Tabular presentation of key data to facilitate comparisonUse of two more tables might facilitate comparisons better. For example, the section on Comments on the numbering and naming of avialan specimens from the Solnhofen Archipelago is useful, but quite long. Perhaps include the data in a table listing all 13 specimens, their current taxonomy, their numbers and correlative 'names' and institution, and include just a couple sentences to justify the new names for the eleventh to thirteenth specimens. If length is an issue then it could be in Supplementary file, but it is of more use to the audience in the main text, i.e. not lost/invisible in the Supplementary file.

As one of the peculiarities of *Archaeopteryx* is the use of the numbers of finds and colloquial names rather than the specimens numbers, even in the scientific literature, we think it to be important to clarify both the numbering and the colloquial names used. Tables with lists of specimens (up to the twelfth specimen) can be found in other publications (e.g. Rauhut et al., 2018), so simply adding three names to these lists seems rather redundant. It is furthermore important to point out that especially the numbering of specimens should be fixed, as current developments of taxonomic identifications or uncertainty make the numbering of *Archaeopteryx* specimens somewhat ambiguous (e.g. the twelfth specimen has been called eleventh specimen in the popular press in some cases, following the identification of the Haarlem specimen as a separate taxon).

Similarly, a table with comparative key measurements for the 12 Archaeopteryx specimens including ratios should also be provided, so demonstrating how similar this specimen is to other urvogel specimens.

Tables, showing the measurements of various are *Archaeopteryx* specimens are already available in Mayr et al., 2007 or Wellnhofer, 2009. However, we agree that for comparison it is useful to show those ratios, where *Alcmonavis* differs from *Archaeopteryx*, including T and p values. These values are shown in Table 2.

7) Discussion on possible flight mode of the fossilThis section is entirely speculative and not well substantiated. Various authors have detailed that flapping flight capabilities are unlikely for Archaeopteryx. The different morphology of the supracoracoideus muscle is just one argument contra flapping flight capabilities of Archaeopteryx, and the anatomy of Archaeopteryx does not show any evidence that this animal was capable of take-off through flapping flight (comparisons with extant pigeons certainly are not appropriate, as the latter have a completely different skeletal morphology than any Jurassic avians). Even though Voeten et al. recently proposed flapping flight capabilities of Archaeopteryx, their arguments based on the thickness of the bone cortex are far from compelling. Likewise, it is not clear why the presence of a tubercle for musculus biceps brachii on the radius would indicate flapping flight capabilities (so much the more as the presence/absence of this feature in other Archaeopteryx specimens is difficult to assess). Given the speculative nature of this interpretation, to include in the abstract a sentence like "Several modifications, especially in muscle attachments related to the downstroke of the wing, indicate a rapid adaptation of the forelimb for active flapping flight in the early evolution of birds" is not justified by the data (so much the more as the discussion is far less definitive).

We are aware that the flight mode of *Archaeopteryx* is still highly debated and many scientists favour a gliding flight as it costs less energy. Being familiar with the environment of the Solnhofen Archipelago, we would like to point out that the habitat of *Archaeopteryx* and Co. was an arid bush land with only a few small trees. Consequently, we find an arboreal life style with passive gliding flight for *Archaeopteryx* as often suggested rather unlikely.

The mentioned studies on the musculature of the pectoral girdle just show that the configuration was different in *Archaeopteryx* compared to modern volant birds (it was probably more similar to that of ostriches). If these differences really prevent flapping movements in *Archaeopteryx* is speculative (ostriches are still able to flap their forelimbs). 3D reconstructions of the shoulder girdle and forelimb of *Archaeopteryx*, based on a synchrotron scan of the well-preserved Thermopolis specimen, indicate that *Archaeopteryx* was actually able to perform a wing stroke (Carney, 2016; Heers and Carney, 2017). Due to different muscle configurations the flight stroke was certainly not as well “optimized” than in modern volant birds, but already possible. In the revised version, we cite a couple of recent studies supporting our case.

Showing the presence of the *tuberculum bicipitale radii* in three specimens of *Archaeopteryx*, we do not know why this character is controversial for *Archaeopteryx*. In other specimens the structure is simply not visible due to the orientation of the radius or overlap of matrix or other bones.

To avoid confusion, we never compared the flight performance of *Archaeopteryx* or *Alcmonavis* with that of pigeons. We cited the relevant study, because it shows, which muscles are active during different phases of a wing stroke in modern volant birds. Using this as an analogue, we hypothesize that the presence of *tuberculum bicipitale radii* indicates a stronger *m. biceps brachii*. We believe that this is related to active movements during flight, because for gliding the wing just need to act as passive airfoil and does not require such modifications in the elbow.

Given the great similarities of the general skeletal architecture of this fragmentary wing to the known Archaeopteryx remains (three unfused metacarpals with large ungual phalanges, similar phalangeal proportions etc.), it is highly unlikely that substantial differences in flight mode to Archaeopteryx existed. The speculative nature of this discussion and interpretation should be addressed.

We already made a very detailed comparison to *Archaeopteryx* regarding the enlarged muscle attachment sites present in *Alcmonavis*. Assuming that these areas are associated with stronger muscles, an improvement of the flight ability is likely. Please note, that we never claimed that the flight mode of *Alcmonavis* is substantially different from *Archaeopteryx* (we apologize if the term “rapid” may has caused this impression).

Applying the logic of the argument, no improvement for the flight ability of *Confuciusornis* is to be expected either, as it has three unfused metacarpals with large ungual phalanges, and more or less similar phalangeal proportions.

8) Biogeographic interpretations.These speculations towards the end of the manuscript should be deleted altogether.

We deleted the Conclusions section.